# Adaptive chunking improves effective working memory capacity in a prefrontal cortex and basal ganglia circuit

**Aneri Soni\*, Michael J Frank\***

Brown University, Providence, United States

## eLife Assessment

This **important** work proposes a neural network model of interactions between the prefrontal cortex and basal ganglia to implement adaptive resource allocation in working memory, where the gating strategies for storage are adjusted by reinforcement learning. Numerical simulations provide **convincing** evidence for the superiority of the model in improving effective capacity, optimizing resource management, and reducing error rates, as well as for its human-like performance. This work will be of broad interest to computational and cognitive neuroscientists, and may also interest machine-learning researchers who seek to develop brain-inspired machine-learning algorithms for memory.

**\*For correspondence:**
aneri_soni@brown.edu (AS);
Michael_Frank@brown.edu (MJF)

**Abstract** How and why is working memory (WM) capacity limited? Traditional cognitive accounts focus either on limitations on the number or items that can be stored (slots models), or loss of precision with increasing load (resource models). Here, we show that a neural network model of prefrontal cortex and basal ganglia can learn to reuse the same prefrontal populations to store multiple items, leading to resource-like constraints within a slot-like system, and inducing a trade-off between quantity and precision of information. Such 'chunking' strategies are adapted as a function of reinforcement learning and WM task demands, mimicking human performance and normative models. Moreover, adaptive performance requires a dynamic range of dopaminergic signals to adjust striatal gating policies, providing a new interpretation of WM difficulties in patient populations such as Parkinson's disease, ADHD, and schizophrenia. These simulations also suggest a computational rather than anatomical limit to WM capacity.

## Introduction

It has long been appreciated that WM capacity is limited, but despite many decades of research, the nature of these limitations remains controversial. For example, early work by *Miller, 1956* famously posited that WM is limited to roughly seven items, but he also stated that the precise limitation might vary depending on the information quantity of the relevant memoranda. More recently, a vigorous debate over the last two decades has divided roughly into two schools of thought. The 'slots' theory argues that WM is limited to a specific number of items. According to this account, each item is stored in a discrete slot and encoded with high precision, leading to low or zero errors when that item is probed at recall. When the number of items to be remembered exceeds the capacity (roughly four items; *Cowan, 2008*), some items will be forgotten, and therefore participants resort to guessing (*Zhang and Luck, 2008*; *Fukuda et al., 2010*; *Luck and Vogel, 2013*). In contrast, the 'resource' theory argues that people can store an arbitrarily large number of items in WM, with no inherent item limit, but that each item competes for a shared pool of resources. As a result, the precision of

each memoranda goes down with each added item to be recalled. In the limit, when many items are presented, each one is recalled with low precision, which can masquerade as guessing (*Bays et al., 2009*; *Ma et al., 2014*).

Critically, regardless of the distinction between discrete and continuous resources, the measured WM 'capacity' from experimental data is not fixed. For example, individual differences in WM capacity are largely determined not by the raw number of items one can store but rather one's ability to filter out distracting items (*Vogel et al., 2005*; *McNab and Klingberg, 2008*; *Astle et al., 2014*; *Feldmann-Wüstefeld and Vogel, 2019*). More generally, one can leverage various (potentially unconscious) memory strategies to improve 'effective capacity,' leading to experimentally observed capacity measurements that fluctuate depending on stimulus complexity, sensory modality, and experience with the stimuli (*Pusch et al., 2023*). Thus a key but often overlooked aspect of WM lies in the efficient *management* of access to and from working memory. Taking into account this management may also provide a mechanism for understanding WM not only in terms of maintenance, but also how gating strategies may be used for manipulation of information – i.e., 'working with memory' (*Moscovitch and Winocur, 2009*). In other words, effective capacity encompasses gating items into/out of WM as well as the maintenance of items.

For example, recent theoretical and empirical work suggested that information stored in WM can be partitioned into discrete representations, but that similar items could be stored in a shared partition, in effect chunking them together (*Nassar et al., 2018*). Intuitively, it is simpler to remember that one needs to purchase dairy items, bread, and fruit rather than to remember to buy milk, cheese, bread, oranges, and bananas. This active chunking strategy serves as a lossy information compression mechanism: it frees up space for other items to be stored and recalled. This happens at the cost of reducing precision for the chunked items. Experimental evidence provides support for such a model over alternatives, and provided a mechanism to explain why precision can be variable across trials as posited by previous resources models (*van den Berg et al., 2012*). Moreover, (*Nassar et al., 2018*) showed that the optimal chunking criterion (i.e. how similar two items need to be to merit storing as a single chunk) varies systematically with set size (number of items to be remembered) and can be acquired via reinforcement learning (RL). In line with this account, they reported evidence that humans adapted chunking on a trial-by-trial basis as a function of reward feedback in their experiment. They also performed a meta-analysis of other visual working memory (VWM) datasets showed that optimal performance was associated with more chunking with increasing set size. Thus, at the cost of small errors (due to loss in precision), normative models and humans have overall better recall and performance when employing this chunking method. This and related theories (*Brady et al., 2011*; *Brady and Alvarez, 2015*; *van den Berg et al., 2014*; *Wei et al., 2012*; *Swan and Wyble, 2014*; *Nassar et al., 2018*) may reconcile differences between the slots and resources theories.

While these normative and algorithmic models are consistent with experimental data, it is unclear how *biological* neural networks could perform such adaptive and flexible chunking. If a biological neural network exhibits the same mechanism, we can make more clear predictions about human WM as well. The dominant neural model of visual working memory is the ring attractor model of prefrontal cortex (PFC), whereby multiple items can be maintained via persistent activity in attractor states (*Edin et al., 2009*; *Wei et al., 2012*; *Nassar et al., 2018*). In these models, nearby attractors coding for overlapping visual stimuli can collide, leading to a form of chunking and loss of precision, and where some items are forgotten due to lateral inhibition (*Wei et al., 2012*; *Almeida et al., 2015*; *Nassar et al., 2018*). While these models have successfully accounted for a range of data, by modeling only the PFC (or a single cortical population), they have limitations. Firstly, these models cannot determine whether or not an item should be stored. In other words, unlike humans (*Vogel et al., 2005*; *McNab and Klingberg, 2008*), they cannot improve effective capacity by filtering content to only include relevant information. Second, any chunking that occurs in these models is obligatory – determined only by how overlapping the neural populations are and hence whether attractors will collide. Thus, chunking can't be adapted with task demands as required by normative models and human data (*Nassar et al., 2018*). Finally, during recall, these network models cannot select a specific item from memory based on a probe (accuracy in these models is considered high as long as the relevant stimulus is encoded somewhere in the pool of neurons; *Edin et al., 2009*; *Wei et al., 2012*; *Almeida et al., 2015*). In other words, these models have no way of manipulating or accessing the contents of the WM.

This selective access and management of information in WM requires the brain to (1) solve the variable binding problem and (2) create a role- addressable memory. Variable binding refers to the ability to link attributes of an object together (for instance a person has a name, face, location etc.) In WM, humans have to temporarily bind a given memorandum with a given slot in memory, often in a role-addressable manner. For example, they might need to recall the color of an object based on its shape. Indeed, limitations in WM capacity have been linked to difficulty in binding items to their respective slots in memory (*Oberauer, 2013*; *Oberauer and Lin, 2017*). Moreover, the selective updating of information in these slots is thought to produce a recency bias ubiquitously observed in human WM (*Oberauer et al., 2012*).

Another complementary line of biologically-inspired neural network models addresses how interactions between basal ganglia (BG), thalamus, and PFC support independent updating of separable PFC 'stripes' (anatomical clusters of interconnected neurons that are isolated from other stripes; *Levitt et al., 1993*; *Pucak et al., 1996*; *Frank et al., 2001*). These prefrontal cortex basal ganglia working memory (PBWM) models focus on the aforementioned 'management' and variable binding problem. They simulate the brain's decision whether to encode a sensory item in WM ('selective input gating'). They also simulate which item (of those stored in WM) should be accessed ('output gating') for reporting or subsequent processing (*O'Reilly and Frank, 2006*; *Hazy et al., 2007*; *Krueger and Dayan, 2009*; *Stocco et al., 2010*; *Frank and Badre, 2012*; *Kriete et al., 2013*). The combination of input and output gating decisions that are made can be summarized as the *gating policy*. Via dopaminergic reinforcement learning signaling in the BG, the networks learn an effective gating policy for a given WM task (*Frank and Badre, 2012*). This policy includes (i) whether or not to store an item (i.e. if it is task-relevant or distracting), (ii) if relevant, in which population of PFC neurons to store it, and (iii) which population of PFC neurons should be gated out during recall or action selection. As such, PBWM networks can perform complex tasks that require keeping track of sequences of events across multiple trials while also ignoring distractors. The PBWM framework also accords with multiple lines of empirical evidence, ranging from neuroimaging to manipulation studies, suggesting that the BG contributes to filtering (input gating) of WM (which improves effective capacity) (*McNab and Klingberg, 2008*; *Cools et al., 2007*; *Cools et al., 2010*; *Baier et al., 2010*; *Nyberg and Eriksson, 2016*) and selecting among items held in WM (output gating; *Chatham et al., 2014*). Evidence also supports the PBWM prediction that striatal DA alters WM gating policies analogous to its impact on motor RL (*O'Reilly and Frank, 2006*; *Moustafa et al., 2008*); for review see *Frank and Fossella, 2011*. Finally, these human studies are complemented by causal manipulations in rodent models implicating both striatum and thalamus as needed to support WM maintenance, gating, and switching (*Rikhye et al., 2018*; *Nakajima et al., 2019*; *Wilhelm et al., 2023*). However, to date, these PBWM models have only been applied to WM tasks with discrete stimuli and thus have not addressed the tradeoff between precision and recall in VWM. Due to the discrete nature of the stimuli, accuracy is typically binary, and WM information could not be adaptively chunked. Furthermore, previous PBWM studies only trained and tested within the allocated capacity of the model, limiting the application to common human situations in which set size of relevant items goes beyond WM capacity.

In sum, these two classes of neural WM models address complementary phenomena but their intersection has not been studied. In particular, how can our understanding of BG-PFC gating inform the slots vs resources debate and the nature of WM capacity limits more generally? On the surface, PBWM is a slots model: it has multiple, isolated PFC 'stripes' that can be independently updated and accessed for read-out. Note, however, that performance is improved in these models when they use distributed representations within the stripes (*Hazy et al., 2007*; *Kriete et al., 2013*), which can have resource constraints (*Frank and Claus, 2006*). We thus considered whether PBWM could acquire, via reinforcement learning, a gating strategy whereby it stores a 'chunked' representation of multiple items within the same stripe, leaving room for other stripes to store other information and *in effect increasing the effective capacity without increasing the allocated capacity*.

Here, we sought to combine successful aspects of both models. We considered whether PBWM could be adapted to perform VWM tasks with continuous stimuli and whether it can learn a gating policy via RL that would support chunking to meet task demands. We include a ring attractor model that allows for mergers of continuous-valued stimuli via overlapping representations (*Wei et al., 2012*; *Edin et al., 2009*; *Almeida et al., 2015*; *Nassar et al., 2018*). But rather than representing all input stimuli at once, the network evaluates a single sensory input in the context of stimuli already stored in

one or more PFC stripes. The ring attractor can then merge or chunk the sensory input with its nearest neighbor in PFC. Importantly, the resulting chunks are not obligatorily stored in WM. Rather, the BG learns a gating policy so that it can potentially store the original (and more precise) sensory input, or it can replace a currently stored representation with the chunk – and adaptively alter its strategy as a function of task demands (set size). During recall, the network can gate out the corresponding (original or chunked) representation linked to the probe, and reproduce hallmarks of human performance in continuous report VWM tasks.

Notably, we find that this chunk-augmented PBWM network outperforms control models that lack chunking abilities across a range of task conditions. Chunk models outperform control networks even when the control models are endowed with an allocated capacity that exceeds the set size. This latter result stems from a credit assignment problem that arises when networks must learn to store and access multiple items in WM. Chunking instead allows for a common set of stripes to be repeatedly reused and reinforced, limiting the number of possible solutions explored. As such, this result lends insight into a normative rationale for why WM capacity is limited in the first place. Moreover, these performance advantages depend on a healthy balance of BG dopamine signaling needed to support adaptive gating policies that enhance effective capacity, providing a novel account for WM deficits resulting from aberrant BG DA signaling in patient populations such as Parkinson's disease and schizophrenia (*Frank, 2005*; *Moustafa et al., 2008*; *Cools, 2006*; *Cools et al., 2010*; *Maia and Frank, 2017*). Finally, we show that, like humans, the model shows a recency bias, with increased accuracy for reporting items that had been presented more recently. This results both from an increased propensity to update items that had been presented earlier and as a consequence of chunking of earlier items. This account is consistent with evidence in the human literature on the nature of recency effects *Oberauer et al., 2012*, and inconsistent with alternative neural models of passive decay.

## Methods

The model is implemented using an updated version of the Leabra framework (*O'Reilly et al., 2024*), written in the Go programming language (see https://github.com/emer/emergent, copy archived at *emer, 2025*). All of the computational models, and the code to perform the analysis, are available and will be published on our GitHub account. We first outline the basic neuronal framework before elaborating on the PBWM implementation, modifications to the continuous report task, and chunking implementation, with most details in the Appendix.

Leabra uses point neurons with excitatory, inhibitory, and leak conductances contributing to an integrated membrane potential, which is then thresholded and transformed to produce a rate code output communicated to other units.

The membrane potential $V_m$ is updated as a function of ionic conductances $g$ with reversal (driving) potentials $E$ according to the following differential equation:

$$C_m \frac{dV_m}{dt} = \begin{aligned} & g_e(t)\bar{g}_e(E_e - V_m)+ \\ & g_i(t)\bar{g}_i(E_i - V_m)+ \\ & g_l(t)\bar{g}_l(E_l - V_m), \end{aligned} \tag{1}$$

where $g_e^\Theta$ is the level of excitatory input conductance that would put the equilibrium membrane potential right at the firing threshold $\Theta$ and depends on the level of inhibition and leak.

where $C_m$ is the membrane capacitance and determines the time constant with which the voltage can change, and subscripts $e$, $l$ and $i$ refer to excitatory, leak, and inhibitory channels, respectively.

The excitatory net input/conductance $g_e(t)$ is computed as the proportion of open excitatory channels as a function of sending activations times the weight values:

$$g_e(t) = \langle x_i * w i \rangle = \frac{1}{n}\sum_i x_i w_i \tag{2}$$

Activation communicated to other cells ($y_j$) is a thresholded ($\Theta$) sigmoidal function of the membrane potential with gain parameter $\gamma$:

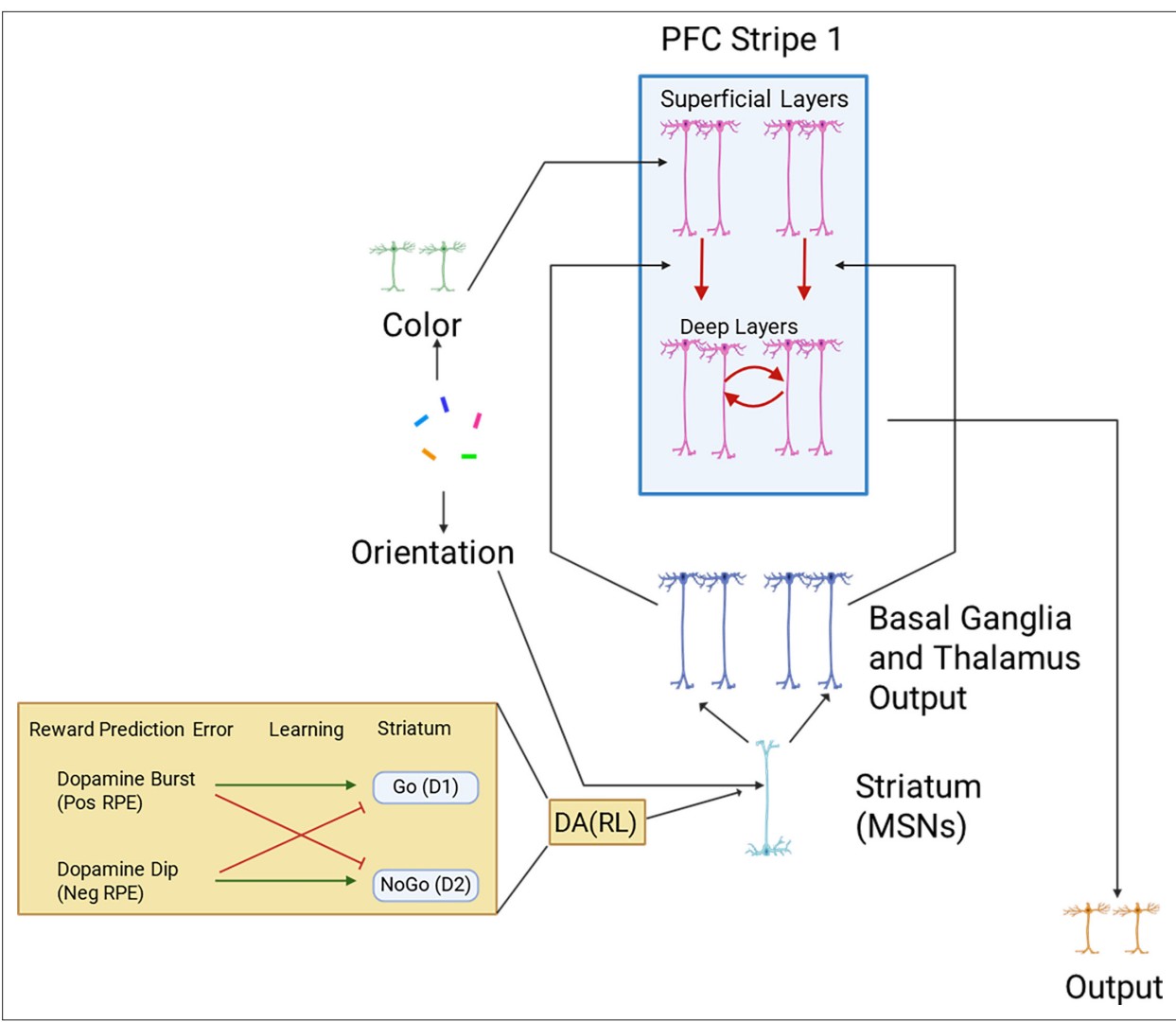

**Figure 1.** Base model sensory inputs (reflecting visual cortical representations) project to PFC superficial layers, which transiently represent those inputs. Activity is maintained in prefrontal cortex (PFC) after stimulus offset (and into subsequent trials) only when gated. Red arrows indicate gating, supporting transfer of information from superficial layers to deep layers, triggered by striatal disinhibition of dorsomedial thalamocortical activity. Maintenance is represented by recursive red arrows in deep layer. Green insert shows how striatum D1 and D2 neural populations (which have opposite effects on gating) are modulated by dopaminergic reward prediction errors (RPEs). Over the course of learning, synaptic weights evolve to support effective gating strategies that increase rewards.

$$y = \frac{1}{\left(1 + \frac{1}{\gamma[g_e - g_e^{\Theta}]_+}\right)}$$

$$g_e^{\Theta} = \frac{g_i(E_i - \Theta) + g_l(E_l - \Theta)}{\Theta - E_e} \tag{4}$$

Further details are in the appendix, but we elaborate the inhibition function below given its relevance for the chunking mechanism.

## Base PBWM model

The PBWM model (*Figure 1*) is built based on a large repertoire of research surrounding WM, gating, and RL and has been developed over a series of articles (see introduction). Here, we focus on the high level description of its functionality and the unique additions to the current application, particularly the implementation of continuous rather than discrete representations throughout the network (input, PFC, and response layers), and the chunking mechanism.

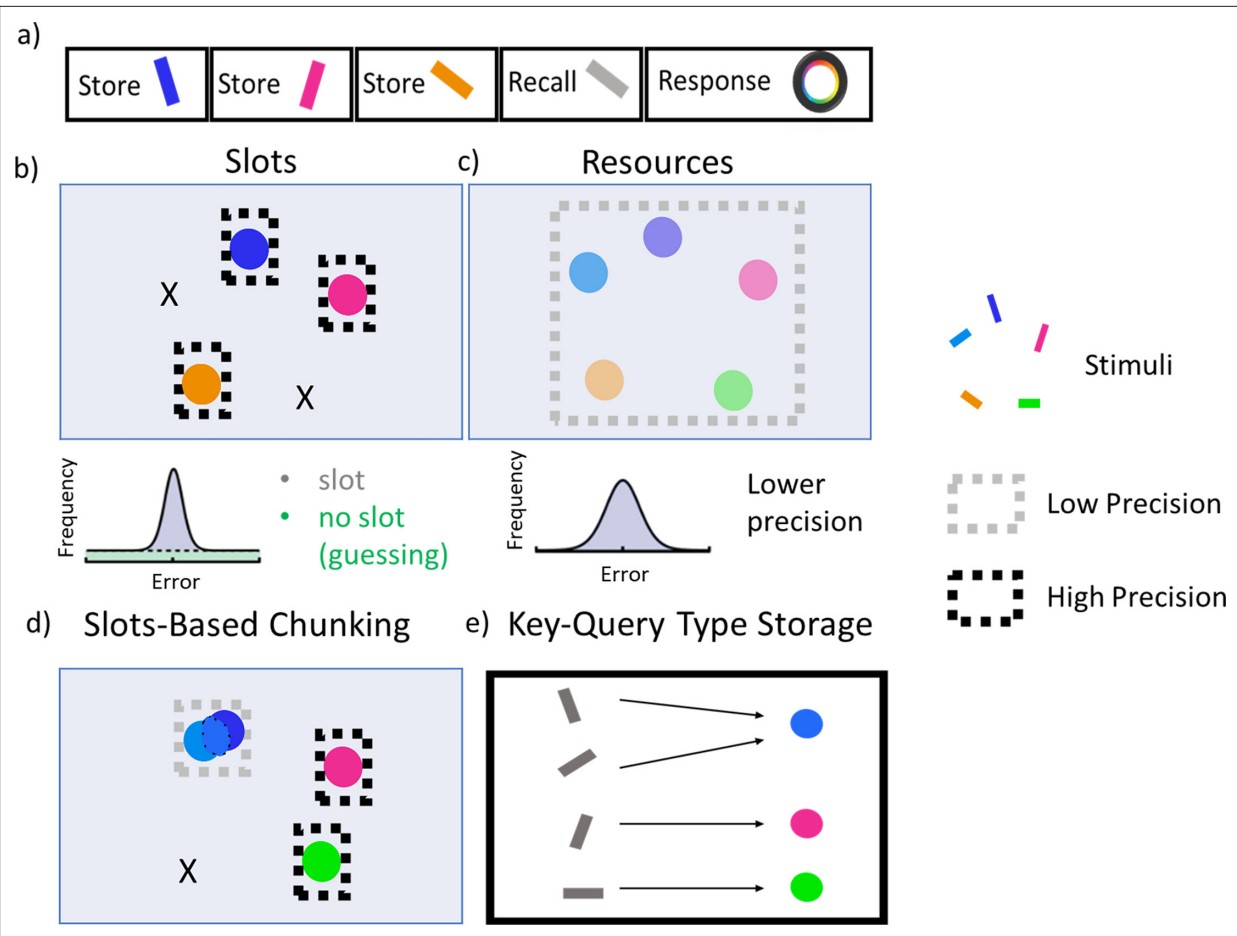

**Figure 2.** Visual working memory task. (**a**) The color wheel task is commonly used to study the nature of capacity limitations in VWM. During encoding, participants are presented multiple randomly generated oriented and colored bars. After a delay they are shown a recall probe trial in which one of the previously seen orientations is presented in gray. The participant responds by using a color wheel in an attempt to reproduce the color associated with that probe orientation. The number of store items are dictated by set size. (**b**) Slots models suggest that WM capacity is limited by a fixed number of slots. When set size exceeds capacity, some items are stored in memory with high precision while the rest are forgotten, resulting in an error histogram that is a mixture of high precision memory (for items in a slot) and guessing (for items not in a slot). (**c**) Resource models state that all items can be stored in a common pool, but as the number of items increase, the precision of each representation decreases, resulting in an error histogram with a large variance (but no guessing). Adapted from *Ma et al., 2014*. (**d**) A hybrid chunking model containing discrete slots, but with resource-like constraints within each slot. Here, the two bluish items are merged together within a slot, reducing their precision but freeing up other slots to represent pink and green items with high precision. The orange item is forgotten. The criterion for chunking can be adapted such that error histograms will look more like the slots theory or resource theory depending on task demands (WM load and chunkability of the stimulus array; *Nassar et al., 2018*). (**e**) Storage in the PBWM-chunk model is like a key-query. The colors are stored as continuous representations in prefrontal cortex (PFC) and can be merged. The orientations are the queries used to probe where information should be stored and where to read it out from.

We modified PBWM to accommodate continuous representations such as those used in the delayed report color wheel task (*van den Berg et al., 2012*; *Nassar et al., 2018*), see *Figure 2a* common task to assess precision and recall characteristics of VWM and which forms the main basis for our simulations. In this task, participants are presented with multiple colored bars on a visual display where each bar has two attributes: a color and orientation. After some delay, participants are shown only one of the orientations in gray, and the subject's task is to report the color that was associated with that orientation using a continuous color wheel. Previous PBWM applications used only discrete stimuli and did not address precision. To simulate continuous report tasks, we represented the color for each stimulus as randomly varying from 0 to 2pi using a population code, where each neuron in the layer maximally responds for a particular color, and the full representation for a given continuous input is represented as a Gaussian bump over an average of 10 neurons in a 20 neuron layer. This representation accords with that seen in visual area V2, with hue neurons that are spatially organized

according to color (*Xiao et al., 2003*). Each color was presented to the network together with a separate representation of its associated orientation (for simplicity we used discrete orientations, as the task is not to recall the precise orientation but only to use it to probe the color). The stimuli are presented sequentially to the mode. This serves three purposes: to simplify the binding problem, to mimic a sequential attentional mechanism, and to make contact with literature on serial WM tasks, (While this sequential presentation of stimuli simplifies the binding problem, it also adds a further challenge as the network must have capacity to recall items that were presented several time steps/trials ago. To solve this problem, the model must learn an efficient and appropriate gating strategy. This also makes the model vulnerable to serial dependence it its chunking, consistent with that observed empirically, whereby WM reports are biased towards the last stimulus that was presented *Bliss et al., 2017*; *Kiyonaga et al., 2017*; *Fischer and Whitney, 2014*).

These input layers project to the PFC maintenance layers, which contain isolated populations in discrete stripes, also coded as Gaussian bumps of activity. As in prior PBWM models, the PFC is divided into superficial and deep layers (*O'Reilly and Frank, 2006*; *Hazy et al., 2007*; *Frank and Badre, 2012*; *Hazy et al., 2021*). The superficial PFC layers for each stripe will always reflect the input stimuli transiently, as candidates to be considered for storage in WM. But for these stimuli to be maintained robustly over delays (and over intervening other stimuli on subsequent time points), they have to be gated into WM. Accordingly, each PFC stripe is modulated by a corresponding BG module consisting of striatal 'Go' and 'NoGo' units which in turn, via direct and indirect pathways, project to BG output/thalamic units. When there is relatively more Go than NoGo activity in a given module, the corresponding Thalamic output unit is activated, inducing thalamocortical

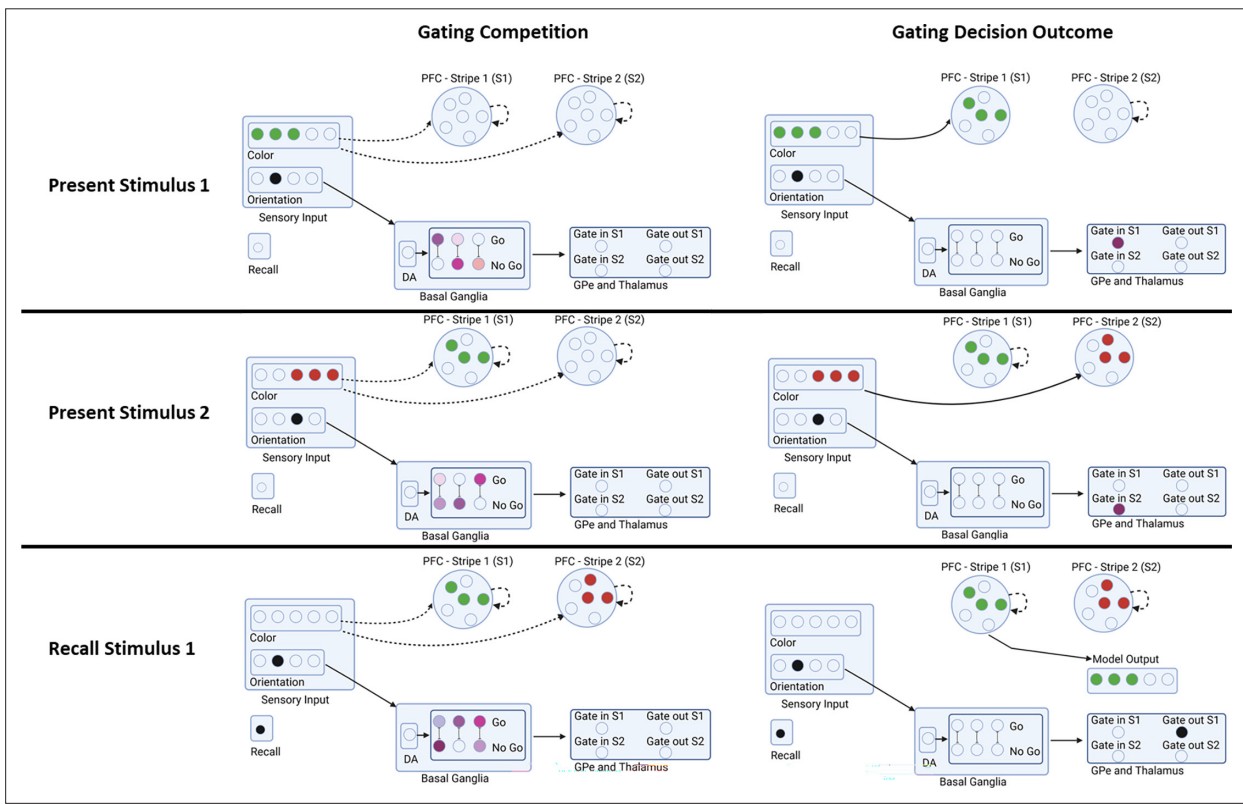

**Figure 3.** Example sequence of network gating decisions. In this example trial, the network is presented with stimulus 1 (color and orientation), stimulus 2, and is then asked to recall the color of stimulus 1 based on just its orientation. Each step is broken into a gating competition that involves the Basal Ganglia (striatal Go/NoGo units, Gpe/Gpi) and Thalamus units. The outcome of this internal competition determines the gating decision and the model output. When the first stimulus is presented, the relative activities determine if and where the stimulus is gated in (stripe 1 or stripe 2). The network gates stimulus 2 in a different stripe based on its orientation. During recall, the network uses a gating policy to output gate the stripe corresponding to the probed orientation. A reward is delivered to the network proportional to the accuracy in reporting the original color. A negative reward is delivered if the color is not sufficiently close (see Methods). Rewards translate into dopaminergic reward prediction error signals that serve to reinforce or punish recent gating operations. This schematic is illustrative; the actual network contains a broader population code and the prefrontal cortex (PFC) stripes are divided into input and output representations, each with deep and superficial layers (see Text).

reverberation and activation of intrinsic ionic maintenance currents, thereby triggering robust maintenance of information in the deep PFC maintenance layers (*O'Reilly and Frank, 2006*; *Hazy et al., 2007*; *Frank and Badre, 2012*; *Hazy et al., 2021*). Thus only the stripes that have been input-gated continue to maintain the most recent color representation in an attractor over time. Importantly, gating in PBWM implements a form of 'role-addressable' memory: the decisions about whether and which stripe to gate colors into depends on its assigned role. In this case, the orientation probe associated with the color determines where it should be gated. By receiving inputs from the orientations, the BG can thus learn a gating policy whereby it consistently stores some orientations into a particular PFC stripe, making it accessible for read out. *Figure 3* shows a schematic example in which, based on the orientation the network gates the first PFC stripe to store the green color, but then stores the color of the second orientation to store the red color. Thus, the PBWM stripes serve a variable binding function (*O'Reilly and Frank, 2006*) which can also be linked to key/query coding (*Traylor et al., 2024*; *Swan and Wyble, 2014*): in this case, the network can learn to use the orientations to guide which stripe is accessed, and the population within the stripe represents the content (color).

During a recall trial, the network is presented with only a single orientation probe. The model needs to correctly 'output gate:' select from multiple PFC maintenance stripes, so that only a single representation is activated in the corresponding PFC 'output stripes,' which in turn projects to the output layer (*Figure 3* bottom row). If it correctly recalls the associated color (by activating the color population in the output layer) it receives a reward. Rewards were given in a continuously linear fashion based on the difference between output and the target color. The activity of the output neurons was decoded (using a weighted linear combination of neuron activities; *Almeida et al., 2015*) and the reward given was inversely proportional with the error. Correctly recalling a color thus requires not only having it stored in PFC, but reading out from the correct stripe that corresponds to the probed item. This read-out operation involves a BG output gating function (*Hazy et al., 2007*; *Frank and Badre, 2012*; *Kriete et al., 2013*), facilitating transmission of information from a PFC maintenance stripe to the corresponding PFC output stripe. In this way, the model can read out from its several stored WM representations according to the current task demands (see *Chatham et al., 2014* for neuroimaging evidence of this BG output gating function). The input and output gating mechanism in PBWM performs a role-addressable gating function that can be linked to the key-query operations in Transformers (*Traylor et al., 2024*).

Note that successful performance in this and other WM tasks (*Hazy et al., 2007*; *Frank and Badre, 2012*) requires learning both the proper input gating strategies (which PFC stripes to gate information into, depending on the orientation, and which to continue to maintain during intervening items), and output gating strategies (which PFC stripes to read out from in response to a particular orientation probe). Such learning is acquired via reinforcement learning mediated by dopaminergic signals projecting to the striatum (represented by the bottom half of the model). At the time of recall, the network receives a dopamine (DA) burst or dip conveying a reward prediction error signal (RPE, by comparing the reward it receives with the reward it expected to receive using a Rescorla Wagner delta rule algorithm). These DA signals are used to modulate synaptic plasticity in the Go and NoGo units, reinforcing corticostriatal Go signals that drive adaptive input and output gating operations (following positive RPEs), and reinforcing NoGo units that suppress ineffective gating strategies (following negative RPEs). (See Appendix for information on parameter searching for other parameters of the model.) The model's previous gating decisions are flagged using an eligibility trace via synaptic tagging. Thus, Rewards modulate Go and NoGo synaptic weights not only based on their current activity levels but also based on these synaptic tags reflecting recent activity.

In addition to the gating strategies, which are learned via dopaminergic reinforcement learning as per above, the network also has to learn to produce the correct color in the output layer. On each Store and Ignore trial, regardless of whether stimuli are gated into PFC, the network has to report the color presented in the input. As such the connections from the input to output layers are plastic, and this mapping is learned via supervised learning (i.e. target colors are presented and the network learns from errors using the XCAL learning rule, see appendix). On recall trials, the network also has to learn to map the neurons that are output gated from PFC to reproduce the associated color in the output layer. As such, connections from PFC output stripes to the output layer are also plastic and learns according to the same rule. (Note that this is only useful if the corresponding stimuli have been gated

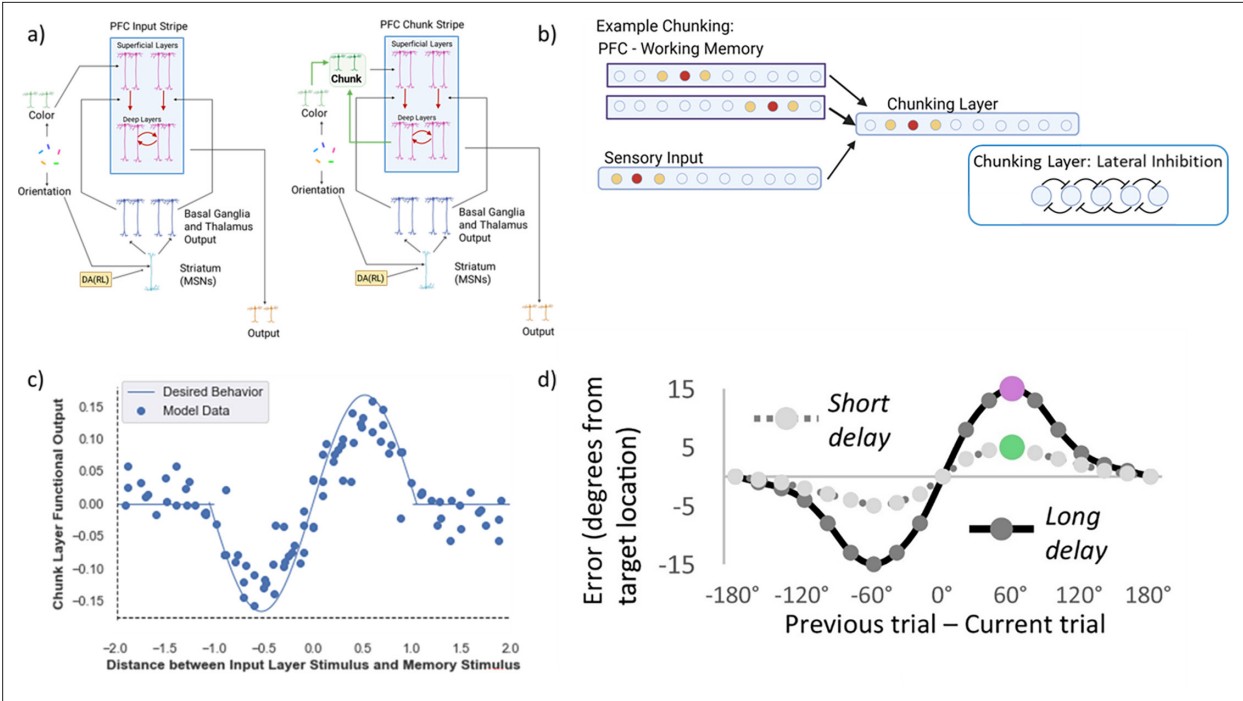

**Figure 4.** Prefrontal cortex basal ganglia working memory (PBWM) model and chunking layer details. (**a**) Network diagram in the minimal case of two stripes: the first prefrontal cortex (PFC) stripe receives projections from the input layer ('PFC Input Stripe'); the second PFC stripe receives projections from the chunk layer ('PFC Chunk Stripe'). The network can be scaled to include more stripes of either type. We will refer to this model as the 'chunk model.' The control 'no chunk' model has two stripes that are both of the type 'PFC Input Stripe' (but it can use them to store separate inputs). (**b**) Chunking schematic. A posterior ring attractor layer receives both bottom up sensory input and top-down input from the two PFC stripes (maintaining separate stimuli/representations). Overlap between the sensory input and the first PFC representation leads to convergent excitatory input in the chunking layer, resulting in a merged attractor. The impact of the more distant PFC representation is suppressed due to lateral inhibition. (**c**) Chunking profile based on similarity. The x-axis shows the difference (in arbitrary units - comparable to radians) between the incoming stimulus and the nearest stimulus in PFC. The y-axis shows the deviation in the decoded chunk layer representation from the input stimulus. If the sensory input is close to a PFC representation, the chunk layer is attracted toward it. If the difference between the input stimulus and the nearest PFC representation is too large, the chunk layer largely mirrors the input (due to stronger input than PFC projections and lateral inhibition). This chunking profile closely matches that seen human memory representations, whereby memory reports are biased toward recent stimuli *Kiyonaga et al., 2017*.

in and out of PFC). All other weights are fixed (i.e. from striatum to GPiThal, and from PFC superficial to deep layers).

All networks are trained for 500 epochs and 100 trials per epoch. This was more than sufficient for learning to converge. We found that despite some fluctuations in training curves late in training, the gating strategy used by the model was largely unchanging at this point.

## Chunking/nearest neighbor implementation

The base model incorporates purely feedforward visual input that can be stored in prefrontal cortex. But it is widely appreciated that there are also top-down projections from prefrontal cortex to posterior sites (e.g. in parietal cortex). We posited that these top-down projections would allow networks to bias the sensory representation toward those stored in PFC, facilitating chunking. The degree of such bias should depend on the strength of those projections, and indeed experimental evidence shows multiple abstractions of an item across parietal cortex and auxiliary sensory regions *Ito and Murray, 2023*.

We considered a minimal set up in which the network has access to two such representations, allowing it to represent both the raw sensory input (through connections directly to PFC) but also via a 'chunking layer' representing posterior cortical association areas that receive excitatory input from both sensory regions and top-down information from deep PFC maintenance layers (*Figure 4a*). This

convergent excitatory connectivity ensures that if any of the currently maintained PFC representations is close enough to the current sensory input, the overlapping neural activations will be enhanced, thereby biasing the bump attractor in the chunk layer to be attracted toward the nearest PFC representation(s). Lateral inhibition ensures that only the most excited units will remain active. As such, the chunk layer will either represent the original incoming stimulus (if no PFC representation is sufficiently close to it) or a combined representation between the incoming and existing stimuli (*Figure 4b*).

More specifically, the excitatory conductance to the chunk layer comes from both the input layer and the PFC and can be summarized in the following equation:

$$g_e(t) = \frac{1}{n} \sum_i x_i w_i \tag{5}$$

where $g_e(t)$ is the excitatory conductance (input) to a layer, $x_i$ is the activity of a particular sending neuron indexed by the subscript i, $w_i$ is the synaptic weight strength that connects sending neuron i to the receiving neuron, and n is the total number of channels of that type (in this case, excitatory) across all synaptic inputs to the unit. Note that the relative strength of projections from input to PFC is scaled, with larger influences of input than PFC (to reflect e.g. number of synapses or proximity to the soma, using a relative weight scale ($w_i$) (see appendix and *Computational Cognitive Neuroscience*, Chapter 2 *O'Reilly et al., 2024* for detailed information)). This allows the chunking layer to preferentially reflect the input, subject to an attraction toward PFC representations that overlap with it (the nearest neighbor; *Figure 4*).

Lateral inhibition regulates activity in the chunking layer and, together with the strength of the top-down PFC projections, determines how close representations must be to bias the input toward the nearest PFC representations. (If the peak of the PFC bump attractor overlaps only with the tail of the sensory input bump, its influence will be largely suppressed due to inhibition). Lateral inhibition is implemented using feedforward (FF) and feedback (FB) inhibition (FFFB) which both alter the inhibitory conductance $g_i(t)$:

$$g_i(t) = \text{Gi} \left[ \text{ff}(t) + \text{fb}(t) \right] \tag{6}$$

where feedforward inhibition ($ff(t)$) is determined by the summed net input to the layer and feedback inhibition ($fb(t)$) is determined by the firing rates of the neurons in that layer, and the $Gi$ gain parameter determines the overall sparsity of the layer (i.e. the relative influence of inhibition compared to excitation $Ge$); see *O'Reilly et al., 2024*.

Critically, a given PFC stripe receives projections from either the sensory input or the chunking layer. (The more general idea is that PFC stripes may have access to posterior cortical layers having varying levels of top-down projections and, therefore, chunking profiles). As such, PBWM could learn a gating strategy to store either the original sensory input into the corresponding PFC stripe or to store the chunked representation into the other stripe. In the latter case, it would replace the existing item stored in that PFC stripe with the new chunked representation, incorporating the novel input stimulus. These gating strategies are not hard-wired but are learned. For instance, the network could learn to use one stripe to store colors linked to particular orientations and use the other stripe for the rest of the orientations, allowing it to appropriately manage where to store and read out information when given the probe. In this case, it would have precise memory for those representations that are stored and accessed, but it would have to guess if the probed item was not stored. At the other extreme, the model could learn to preferentially gate representations into and out of the chunk stripe but with less specificity. We will see later how the model gating policy depends on task demands (specifically set size) and evolves with learning.

For each experiment, at least 80 separate random seeds were run for each network, and results are averaged across them. (For select analyses requiring more observations, 160 separate random seeds were used). To test how set size affects learning and memory performance, the models were trained and tested with set sizes 2, 3, or 4. The set size determines the maximum number of stimuli that can be presented before recall trials. For example, set size 4 means that networks have to maintain up to four items before receiving a recall probe, and it may have to recall any of the preceding items.

## Results

We focus our simulations on variants of the color wheel task (see Methods). Briefly, networks were presented with continuous sensory inputs represented as coarse-coded Gaussian bumps of neural

activity in conjunction with their associated discrete orientation. The number of stimuli to store ('set size') before a recall trial varied between two and four items.

During a recall 'probe' trial, a single orientation was presented to the network, with no color (as in the empirical versions of this task). The model had to reproduce the associated color in the output layer, in the form of a continuous population-coded response. The error is then simply the difference between the decoded color from this population and the ground truth value that was presented during the earlier store trial for that orientation. 'Correct' responses show as data points when errors are close to 0 degrees. If the model outputs an incorrect color, but that color corresponds to a different orientation, this is referred to as a binding or swap error (*Bays et al., 2009*). Finally, if the response is incorrect and nowhere near any of the colors for the stored orientations, it would be referred to as a guess. A guess could land anywhere along the axis of –180–180 degrees and as such manifests as a uniform distribution across trials. If the model produced a non-response (nothing was gated out, e.g. if a stripe was empty), we randomly sampled from the uniform distribution (*Almeida et al., 2015* followed a similar process), mimicking random guessing. These errors (correct responses, guesses, binding errors) manifest themselves in the error distribution.

For proof of concept, we began with a minimal set-up in which all models were allocated two PFC maintenance stripes (we relax this assumption later to compare to models with larger allocated capacity). For all simulations, we compare performance (error distributions, gating strategies, influence of dopamine manipulations) between networks with chunking ability (the 'chunk model') against those with equivalent (or larger) number of stripes. We refer to the 'allocated capacity' as the number of stripes given to the no-chunk model, because this is a hard limit on the maximum number of representations that can be stored and accessed. We refer to the 'effective capacity' as the potentially larger number of items that can be accessed due to an efficient gating policy. Effective capacity can be improved if the network learns to consistently store (input gate) colors of distinct orientations in distinct stripes, and to appropriately read out from (output gate) the corresponding stripe at recall trial. It can also potentially be improved via chunking by increasing the number of representations that can be stored and accessed, (Note however, that effective capacity is not always larger than allocated

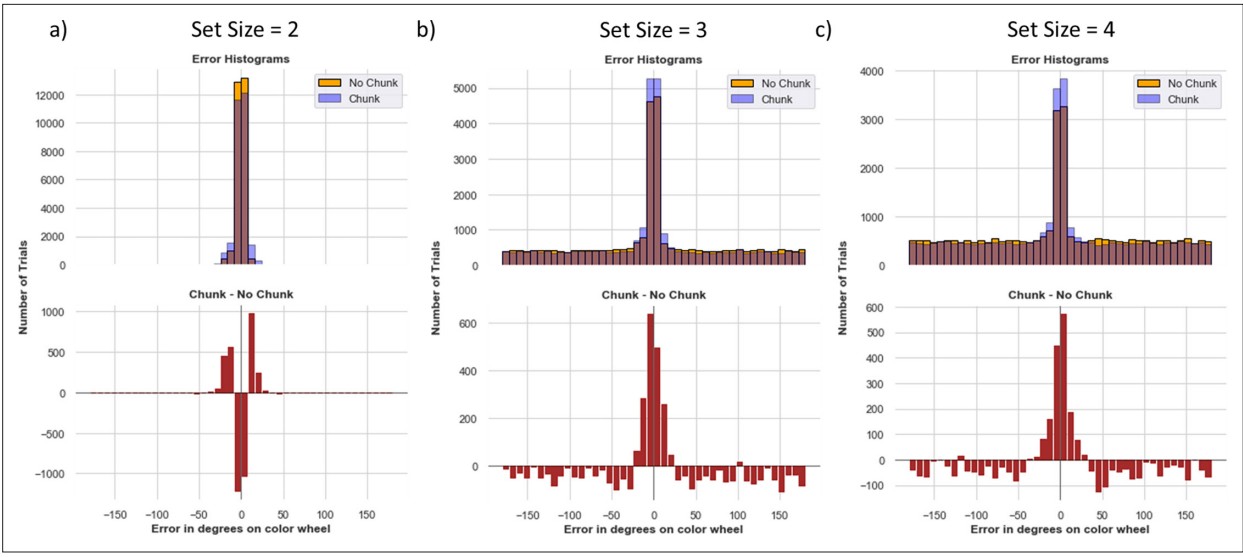

**Figure 5.** Model recall error histograms. The binned error in degrees is plotted on the x-axis, and the number of trials for that error bin on the y-axis. The blue and orange histograms show errors from all recall trials across all 80 random weight initializations from the chunk and no chunk models, each allocated with two stripes. The red histogram plots a bin-by-bin difference in errors between the models. (**a**) For set size 2, there is very little difference between the models. The chunk model exhibits slightly higher rates of low errors neighboring zero (up to 30 degrees), due to small losses in precision resulting from some chunking (see text). (**b**) Set size 3 is beyond the number of stripes allotted to the network. The chunk model has a larger density at zero and small errors, and less guessing (reduced density in the uniform distribution, see red lines). (**c**) At set size 4, the chunking advantage is manifest by low errors and the improvement in less guessing becomes more pronounced (note y-axis scale - the reduction in guessing is actually reduced for set size 4 compared to 3).

The online version of this article includes the following figure supplement(s) for figure 5:

**Figure supplement 1.** P(Recall) across set size.

capacity: without learning an effective input and output gating policy, a network's effective capacity will be less than its allocated capacity, for example if it overwrites information in the same stripe, if it accesses the incorrect stripe during recall, or if it doesn't gate a stimulus at all). It is thus important to note that improving effective capacity requires an effective learning process to develop adaptive gating strategies, as the networks are not hard-coded to use any stripe for storing or accessing any representation. We will show how such learning depends on a healthy dynamic range of dopaminergic RL signals in the basal ganglia.

### Error distributions across set sizes mirror those in human VWM and show chunking advantages

*Figure 5* shows a histogram of network errors (in degrees) during recall trials. The comparisons are made between chunk and no-chunk models as well as set sizes 2, 3, and 4; in these simulations we begin with a minimal setup in which both networks are endowed with two stripes. When set size is equal to the number of stripes (2), errors are small and centered around 0, with some variance due to precision. The overall performance is similar between models, but note that the chunk model shows somewhat more imprecise responses as indicated by some more small error trials. The ability to chunk results in a small cost which manifests as a decrease in precision when chunking occurred but did not need to be used (Note that the chunk model is endowed with two stripes, and thus the only way for it to recall both items is to use both the input stripe and the chunk stripe. As a result, at least one item could be stored precisely in the input stripe, but if the other item is close enough to it, the PFC will store the less precise chunked representation in the chunk stripe, and memory reports will be biased. The network can nevertheless store two precise items if they are far enough apart such that the chunk layer is not biased by the other PFC representation see *Figure 4*).

As set size increases beyond the allocated capacity, both models resort to guessing (random errors) on a subset of trials. This pattern is observable by the error histogram containing a mixture of errors centered around zero and a uniform distribution (see *Figure 1*), as commonly observed in humans (*Zhang and Luck, 2008*). Notably, the chunking model guesses less than the no chunk model and has a higher chance of precisely recalling the items (*Figure 5*). The difference between the chunk and no chunk model widens as the item limit continues to grow beyond the number of stripes. Comparing chunking and non-chunking models illustrates how the benefit of chunking is task-dependent. We next explored how chunking may improve the effective capacity of the network beyond the allocated

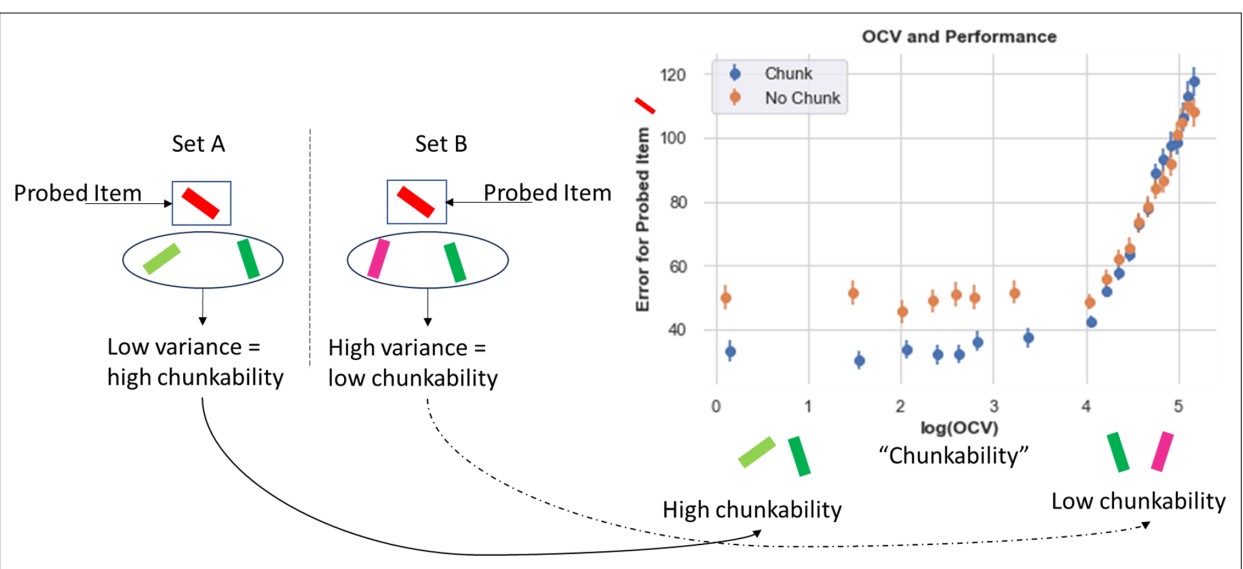

**Figure 6.** Chunking improves recall for non-chunked items. Left. Example array. Here, we compare two sets, both containing a red item that will be later probed. In Set A, the other items (out of the probed cluster) are two shades of green and thus low variance (are similar to each other) and are, therefore, more likely to be chunked. In set B, the out of cluster variance (OCV) for the green and pink items is higher and these items are not likely to be chunked. Right. Chunking networks show consistent Recall advantages (lower errors) when OCV is low and hence the other items are chunkable. This difference disappears as OCV increases and overall errors rise. Errors plotted over all trials averaged over 80 networks in each case.

number of stripes, and more specifically in which trials the advantage manifests, motivated by similar analysis in humans (*Nassar et al., 2018*). We subsequently explore how these strategies are acquired via reinforcement learning.

## Chunking frees up space for other items

A key normative motivation for chunking is that doing so can save space to store other items, thereby improving effective capacity (*Nassar et al., 2018*). In the model, when two items are chunked into one stripe, the second stripe is free to hold another item. One key prediction is that chunking should not only manifest in terms of loss of precision when probed with any of the chunked items, but improved memory for the other items (*Figure 2*). Consider the situation in *Figure 6* left, for a set size of 3. In both Set A and Set B, the red item is the probed item (the one that the model is asked to recall). Set A contains other items in the set that are different shades of green (low variance) and thus 'chunkable.' If the network chunks them together, they will occupy one stripe and the second stripe will be free for the red item, which is then more likely to be recalled at high precision (as it is not part of the chunk). In Set B, the other items are pink and green and are thus not chunkable (high variance). The network may store each of these into a separate stripe, forcing it to randomly guess when probed with the red stimulus. (Alternatively, the red item could be chunked with the pink item, in which case it will be recalled but with less precision than if it stored only the original red item). To formalize and test this intuition, we quantified the 'out of cluster variance (OCV)' as the variance in degrees between the 'other' items in the set (i.e. the items that are not probed; see Appendix for details on OCV calculation). When this variance is low, those items are more chunkable. We then assessed whether accuracy on recalling the probed item (not part of this chunk) is improved as a proxy for chunking behavior.

Indeed, the data supports this prediction (*Figure 6*). When other items in a set are chunkable (low OCV), the chunking network exhibits significantly smaller errors than the control network on the probed item. After some OCV threshold, the chunking network no longer exhibits an advantage, and both networks show increasing errors. (The increasing errors likely result from 'swap errors' *Bays et al., 2009*, i.e., when the network reports one of the other stimuli in its memory instead of the probed item - this results in larger errors as the entire set is more uniformly spaced and thus not chunkable.) In sum, this analysis confirms that chunking does not merely result in reporting imprecise responses for nearby items due to perceptual similarity, but the network leverages such similarity to its advantage so that it can save space for storing and recalling other items, as also seen in humans (*Nassar et al., 2018*).

## Chunking leads to better resource management

In addition to overall better performance, we hypothesized that chunk networks can manage memory resources more efficiently than the no-chunk control models. We next compare how the models use the allocated resources, focusing on store trials in which the maximum number of stimuli were presented (e.g. four stimuli for set size 4). We begin by analyzing the performance of networks after

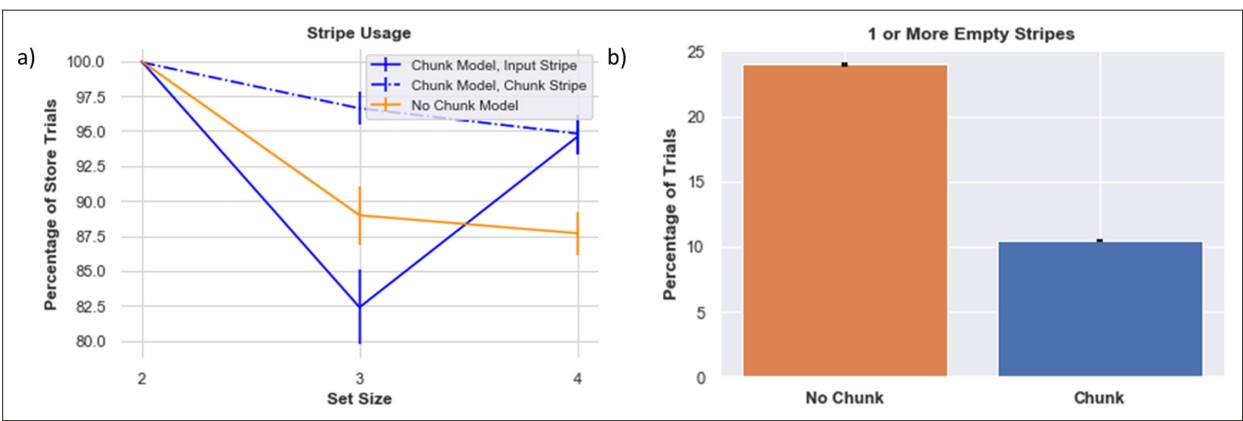

**Figure 7.** Stripe usage. (**a**) Stripe usage for the (1) chunk model, chunk -linked stripe (2) chunk model, input-linked stripe (3) no chunk model (average across both stripes), (**b**) Proportion of trials when at least one stripe was empty. This analysis was done over 160 different model initializations.

learning; below we explore how the chunk network learns to use the different gating strategies over the course of learning for different set sizes.

When the set size is 2, after learning, chunk and no-chunk models are equally able to utilize both stripes on 100% of the trials (*Figure 7*). The networks can properly gate both colors into distinct memory slots without overwriting or reusing the same stripe (in which case *Figure 7* would have shown reduced use of one or the other stripe).

As the set size increases beyond allocated capacity, the gating management problem becomes much harder to solve via RL, and overall performance declines, particularly in the no-chunk (control) model. Indeed, one might expect that as the number of items are increased, the model should always 'max out' the number of stripes used, but in fact the opposite is the case. When the network attempts to gate information into both stripes, the no-chunk model will often receive negative reward prediction errors during recall trials when it will inevitably be forced to guess for a subset of the stimuli that is not in its allocated memory. As a result, input-gating strategies will be punished, even if they were successfully employed and would have been useful had the other stimuli been probed. In turn, due to punishing gating policies that are generally useful, the stripe usage actually decreases, and the stripes are sometimes empty, akin to 'giving up.' Conversely, if the network happened to be positively reinforced for gating a particular stripe, it might promiscuously gate that same stripe for other stimuli. This leads to overwriting information even though the other stripe is empty, and forcing the network to guess (or emit a non-response) when probed with a stimulus that was overwritten. In sum, the model exhibits a non-monotonic use of its resource, as its effective capacity actually declines relative to its allocated capacity. This result is reminiscent of experimental data in fMRI and EEG showing that PFC activity increases with increasing set size but then plummets when set size exceeds the participants capacity (e.g. *Zhang et al., 2016*), perhaps indicating a 'giving up' strategy. We will explore the dopaminergic RL basis of such giving up in a section below.

In contrast, the chunk model is more effective at managing its resources when set size exceeds allocated capacity (*Figure 7*). Recall that the chunk model has access to one stripe with input from the chunked representation ('chunk stripe') and one stripe with raw sensory input ('input stripe'). As set size increases, the network has more opportunities for chunking, and accordingly, relatively more instances of reinforcing the gating operation linked to the chunk stripe. Interestingly, as set size just exceeds allocated capacity (here, for set size 3), the network decreases its use of the input stripe. This pattern arises because the network learns an advantage of chunking for set size 3 and thus sometimes does so more than it needs to (freeing up the input stripe), but also because of the cost of relying too much on the input stripe (as per the no-chunk model). Finally, as set size increases further (set size 4), the chunk network learns the benefit of storing some items in the held out input stripe, increasing effective capacity, while still effectively using the chunk stripe.

In sum, this analysis suggests that resource management and utilization is more consequential than the absolute number of stripes available. In previous network models of visual WM (*Wei et al., 2012*; *Nassar et al., 2018*; *Edin et al., 2009*; *Almeida et al., 2015*), responses were considered 'correct' if the probed item was stored somewhere within the network; i.e., networks were not required to select among these stored representations in response to a probe, and they also did not have to decide whether or not to store an item in the first place. In contrast, the PBWM model focuses on the control of gating strategies into and out of WM, but requires RL to do so. Errors can result from reading out the wrong stripe (a swap error) or from an empty stripe (leading to guessing or non-responding). Chunking is an information compression mechanism that allows multiple stimuli to be mapped onto the same stripe. The chunk stripe has the advantage of being used repeatedly, giving the network has more opportunities to learn how and when to use that stripe.

## Chunking advantages remain even when comparing to networks with higher allocated capacity

One might think that chunking advantages are limited to situations in which the allocated capacity is less than the set size. But when considering the challenges imposed in networks for which storage and access of items is not hard-wired, but must be learned, this is not a foregone conclusion. Indeed, above we found that the number of stimuli stored by no-chunk networks was even lower than their allocated capacity, due to RL challenges. We reasoned that such challenges could persist even when allocated capacity is increased to include or exceed the set size, due to credit assignment challenges

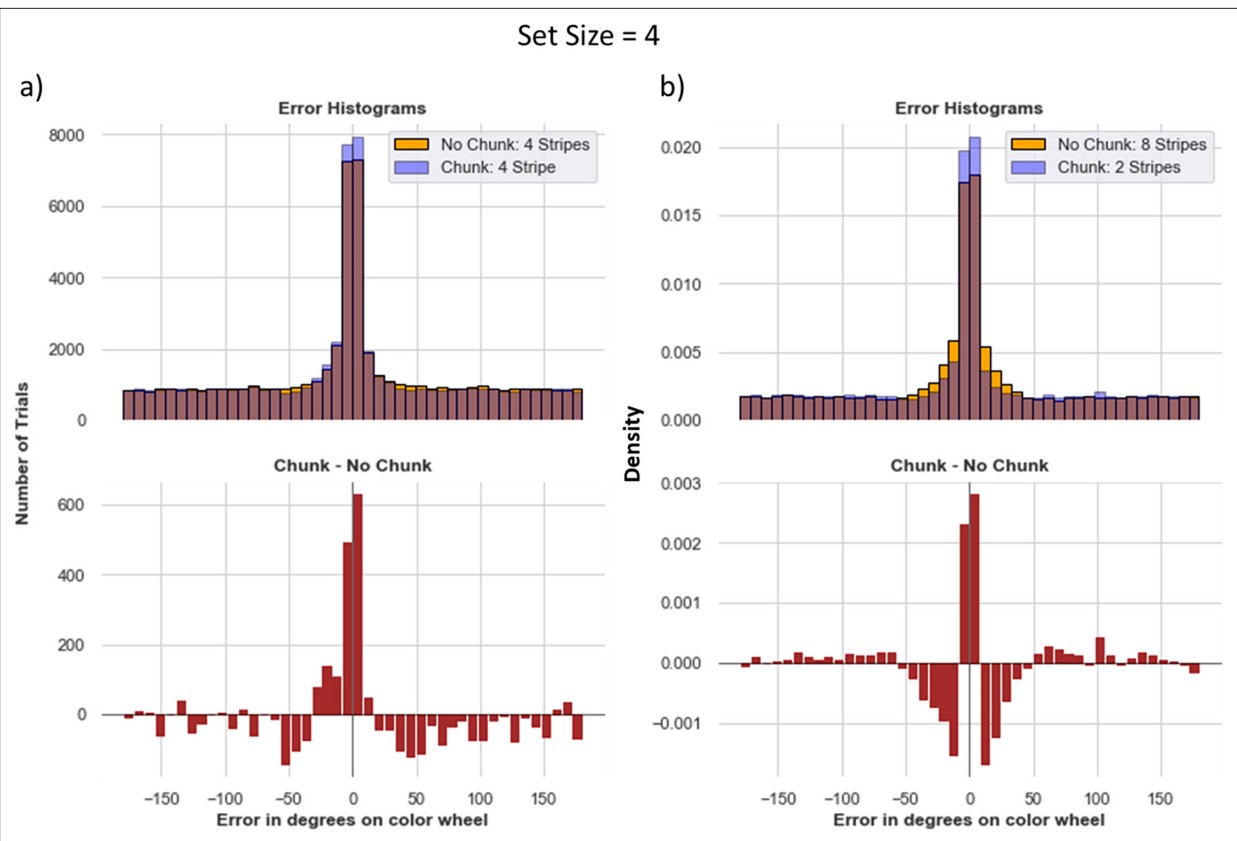

**Figure 8.** Increasing allocated capacity ≠ better performance. The importance of Resource Management (**a**) Chunk model with four stripes vs. No chunk model with four stripes in a task with set size 4. Even though the no-chunk networks has sufficient number of stripes to store each item with high precision, the corresponding chunk network still exhibits advantages, due to difficulties in credit assignment associated with managing four independent stripes. (**b**) A more extreme comparison between the chunk model with two stripes vs. No chunk model with eightstripes. The chunk model guesses slightly more, but has more precise responses. The 8-stripe model has more density at small nonzero errors (see text for explanation). For both a and b the averages were computed over 160 models. For b, we display density rather than counts, because trials where either models gave no response were removed to better understand the small nonzero errors in the eight stripe model (nonresponses add noise).

in networks that are required to learn gating strategies into and out of multiple independent PFC stripes.

We begin with an analysis of networks performing the most difficult task (set size 4) but now allocated four stripes (for both chunk and non-chunk networks; in this case the chunk network has just one chunk stripe and three input stripes). The naïve hypothesis states that no-chunk and chunk models should not differ in performance, or that chunk models might even show a deficit due to loss of precision when using the chunk stripe. Instead, based on earlier results and the above, we reasoned that chunk models might continue to show an advantage here, because frequent reinforcement of gating into and out of the chunk stripe would reduce the credit assignment burden during learning that arises from learning to manage access of four different items into and out of WM.

Indeed, results supported these conclusions. Error distributions in chunk networks have more density at zero and low errors. The chunking model also guesses less (*Figure 8*).

This result largely persisted even in the more extreme case when allocating the control (no-chunk) model 8 stripes (twice as many as needed) and reverting the chunk model to using only two stripes. While guessing is slightly reduced when having more stripes (due to more opportunities to store stimuli), the 8-stripe model does not increase the number of times it precisely recalls the correct item relative to the chunk model with only two stripes (*Figure 8b*). Upon further inspection, one can see that the 8-stripe model produced a larger density of moderate errors around the 0–30 degree range. This result is curious because this model has no ability to chunk. To disentangle the source of these errors and to lend insight into the difficulty of WM gating strategy with high load and/ or allocated capacity, we first removed trials in which the network did not give any output (these

non-responses comprised roughly 12% and 24% of trials from the 8-stripe model and the 2-stripe chunk model, respectively). Within the remaining trials, the 2-stripe chunk model has a higher peak of precise responses (around 0 error) but still slightly more guessing than the 8 stripe no-chunk model, which continues to show a higher density at moderate errors (0–30 degrees). The reason for these moderate errors is that with eight stripes, the network has a more difficult job. It needs to reinforce output gating strategies that properly read out from only the most appropriate stripe. Due to the curse of dimensionality (i.e. when the network outputs a response that is close enough to the target, it will get reinforced for any gating operations that preceded it, leading to spread of credit assignment to other stripes). Indeed, we found that the over-extended eight stripe network frequently reads out from (output gates) multiple stripes in parallel (an average of 2.53 stripes during recall), and thus even when the response does reflect the correct item it is also 'contaminated' by reading out one of the other stripes, such that the averaged response results in a larger number of moderate errors. In contrast, the two stripe networks (across both no chunk and chunk) output gated an average of 1.08 stripes for each trial, appropriately reading out from a single PFC representation.

In sum, simply allocating a network with larger numbers of stripes does not yield the naïve advantages one might expect, at least when gating strategies need to be learned rather than hard-wired. In this case, the networks do use all the stripes available, but don't use them effectively. For example, qualitative observations revealed that a given network might gate one single stimulus into multiple stripes, and then proceed to overwrite many or all the same stripes with a new incoming stimulus – a strategy that is sometimes effective if it happens to get probed for the one of the items still in memory during recall. The large number of input and output gating operations to consider in tandem needed for adaptive behavior leads to a vexing credit assignment problem, as it becomes a challenge to know which of several gating operations or several PFC representations are responsible for task success/error, and networks fall into an unfortunate local minimum. This credit assignment problem is mitigated by chunking, allowing the network to reinforce the same input and output gating policy across multiple instances.

## Frontostriatal chunking gating policy is optimized via RL as a function of task demands

The above results show that chunking networks confer advantages as set size grows, even compared to no-chunk networks that have a larger number of allocated stripes. Moreover, these advantages come with little cost when set sizes are lower (e.g. 2). To explore how the network can adaptively learn to chunk as a function of task demands, we quantified the evolution over learning of each network's 'gating policy.' Prior work has shown that PBWM develops a gating policy that predicts rapid improvement in task success when such policies mimic the task structure e.g. for hierarchical tasks *Frank and Badre, 2012*; see also *Traylor et al., 2024* who showed that modern transformer neural networks mimic a PBWM gating policy when challenged with WM tasks. Here, we assessed whether networks could adaptively learn a gating policy that prioritizes gating into chunk vs input stripes depending on task demands.

In PBWM and related networks, the gating policy is dictated by learned synaptic weights into striatal GABAergic medium spiny neurons (MSNs). These MSNs are classically divided into D1 'Go' MSNs and D2 'NoGo' MSNs, with opponency between these populations determining which actions are selected (i.e. those with the largest difference in Go vs NoGo activities; *Frank, 2005*; *Jaskir and Frank, 2023*). In the case of working memory, a network will be more likely to gate information into a particular stripe if the synaptic weights are larger for the Go in comparison to the NoGo neurons. The relative weights control the disinhibition of that particular stripe. When the network performs well and it gets a reward prediction error, dopaminergic signals modify plasticity into the corresponding D1 MSNs, reinforcing the gating policy that drove the cortical update. Conversely, errors associated with negative prediction errors lead to punishment of that gating policy by increasing synaptic weights into the D2 MSNs (*Frank, 2005*; *O'Reilly and Frank, 2006*). Below we confirm a key role for these dopaminergic signals in modulating adaptive performance. But first here we evaluated how they served to alter specific gating policies. We assessed PBWM gating policies in terms of the differences in Go vs NoGo synaptic weights for each stripe, and how they evolved over time when networks were trained for each set size. Specifically, we computed the summed synaptic connection strengths from the

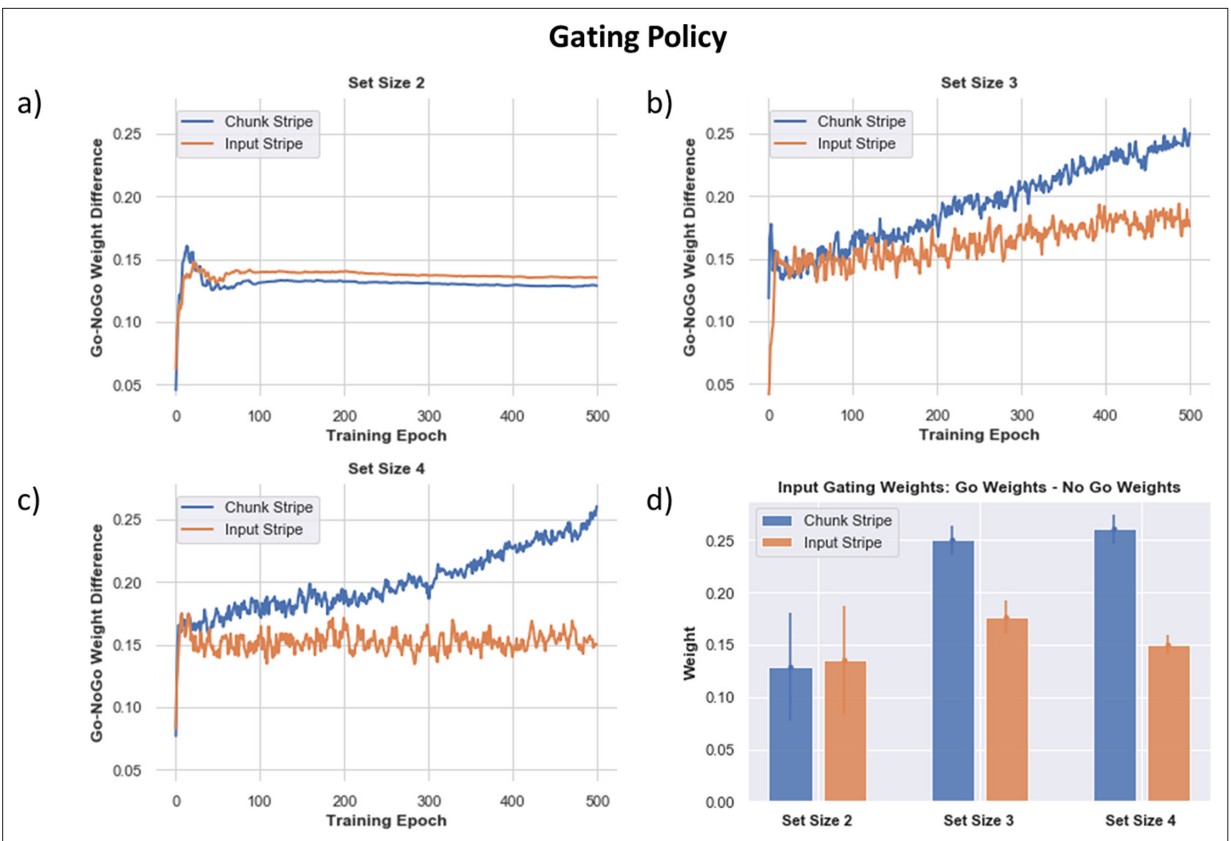

**Figure 9.** Gating policy (Go - NoGo Weights for Each PFC Stripe) across training as the networks learn (over 500 training epochs, averaged over 80 networks), the learned gating strategy differentiates between the input-linked (orange) or chunk-linked (blue) stripes. Positive values indicate the networks learn greater Go than NoGo weights for input gating stimuli into the corresponding stripe. (**a**) Set size 2, the learned gating strategy shows a slight preference for the input stripe to be used (associated with increased precision), but the network also uses its chunk stripe to store the other stimulus (it is possible the chunk stripe stores a merged representation depending on the proximity of the stimuli). (**b**) As the set size increases to 3, the chunk stripe is increasingly preferred over training. (**c**) This differentiation occurs earlier and more strongly for set size 4, where chunking has yet a larger advantage. (**d**) Summary of Go - NoGo weights after training. A larger positive value shows a stronger preference for gating into that stripe. As set size increases, preference for gating into the chunk stripe increases. Relevant for training of all models: We can confirm that the network behavior has stabilized in learning even if the Go/NoGo weights continue to grow over time for the chunked layer (due to imperfect performance and reinforcement of the chunk gating strategy).

Control Input Units representing Store Orientations to the Go and NoGo input gating units in the PFC stripes corresponding to input or chunk:

$$GatingPolicy = \alpha_j = \left[ \frac{\sum Go - \sum NoGo}{\sum Go + \sum NoGo} \right]_+ \qquad (7)$$

(Here $[]_+$ indicates that only the positive part is taken; when there is less Go than NoGo, the net input to the Thalamus is 0).

If the network learns that gating information into the chunk stripe is useful, it will evolve stronger Go vs NoGo weights for that particular gating action. But if instead, it is more useful to gate the veridical input stimulus, it will develop a stronger gating policy for that stripe.

*Figure 9* shows how such gating policies evolve over time. At set size 2 - where allocated capacity (number of stripes) equals task demands, the gating policy slightly prefers to gate into the input stripe. This policy is sensible since the input stripe represents the original stimulus without any loss in precision, yielding lower errors. The network still learns a positive Go-NoGo gating policy for the chunk stripe, because it can use that to represent the other stimulus. Notably, as set size increases, the chunk stripe increasingly becomes the preferred stripe for input gating over the course of learning. This adaptive change in gating policy allows the model to optimize a tradeoff between recall quantity and precision with increasing WM load, mediating the performance advantages described above.

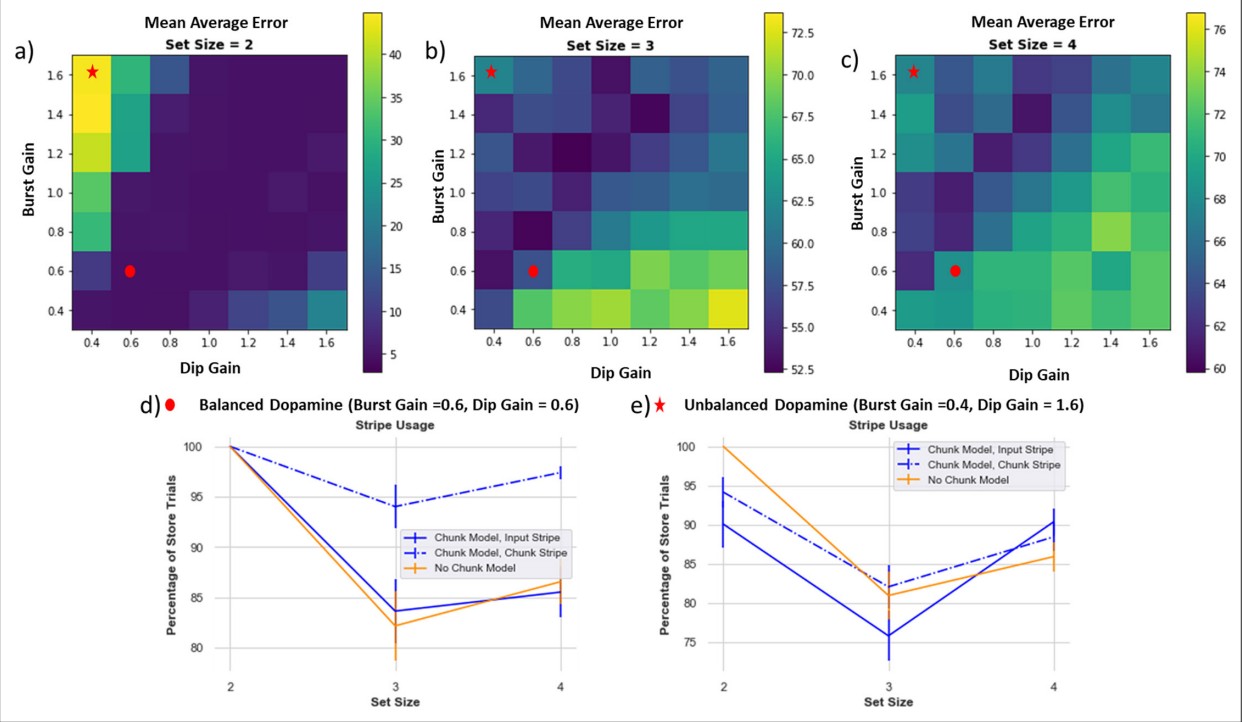

**Figure 10.** Dynamic dopamine bursts and dips are needed for adaptive performance. Each box is an average absolute error over 80 models. The color bar on the right indicates performance (note different scales on each plot), with darker colors (blue) representing better performance/lower absolute error. (**a**) Set size 2: best performance is across the axis where burst and dip gain are symmetrical. (**b** and **c**) Set size 3 and 4: best performance is where burst gain is slightly higher than dip gain. (**d**) Example of balanced DA (burst gain = dip gain=0.6), stripe usage. The chunk model manages to use the chunk stripe across all set sizes and both stripes in set size 2. The no-chunk model shows diminished storage of both stripes with increased set size due to greater propensity of dopamine (DA) dips. (**e**) A regime of DA imbalance (larger DA dip than gain). The chunk model fails to robustly use both of its stripes, losing its advantage. The reinforcement learning (RL) parameters interact with the ability for the chunk model to properly leverage chunking.

These results also accord with those observed in humans by *Nassar et al., 2018*, whereby chunking propensities evolved adaptively as a function of reward history in their experiment, and also in their meta-analysis showing that chunking benefited performance more robustly in experiments with larger set sizes.

## Dopamine balance is critical to learning optimized gating strategies; implications for patient populations

As noted above, learning gating strategies in PBWM is dependent on the basal ganglia and dopaminergic reinforcement system. Both chunk and no-chunk networks must learn whether to gate items into WM, which stripes to gate them into so that they can be later accessed (leaving maintained information in other stripes unperturbed), and during recall, which stripe should be gated out (depending on the probed orientation). To learn this, the network uses a simple RL 'critic' which computes reward expectations and deviations thereof in the form of reward prediction errors (RPEs). Positive RPEs are signaled by dopamine bursts which reinforce activity-dependent plasticity in striatal Go neurons corresponding to recent gating operations (see Appendix and *O'Reilly and Frank, 2006* for details). Conversely, when the model receives a reward that is worse than expected (i.e. it reports an error), a dopamine dip (a decrease in phasic dopamine) will punish previous decisions. This negative RPE will punish the gating decisions by reinforcing corresponding NoGo neurons. To assess whether a healthy balance of such dopaminergic signals are needed for adaptive gating, we manipulated the gains of these dopaminergic bursts or dips to modulate their impact on Go and NoGo Learning. These investigations are relevant for assessing the basic mechanisms of the model but may also have implications for understanding well-documented working memory impairments in patients with altered striatal dopaminergic signaling, such as Parkinson's disease, ADHD and schizophrenia (*Maia and Frank, 2017*; *Cools, 2006*; *Cools et al., 2007*; *Cools et al., 2008*).

*Figure 10* shows how average absolute performance across 80 networks changes with DA burst and dip gain parameters. Overall, a healthy balance of relatively symmetrical DA bursts and dips is needed for optimized performance, but this effect also interacts with set size. The best performance for set size 2 (*Figure 10a*) is along the axis where burst and dip gain are symmetrical. As set size increases, the task becomes harder, and rewards are sparser due to more errors. In this case the best performance is on the axis where burst gain is somewhat greater than the dip gain; the model learns best when it can emphasize learning from sparse rewards.

## Dopamine reinforcement signals can also lead to 'giving up' and diminish effective capacity

When set size increases, learning the proper gating strategies becomes difficult. The models may correctly gate a few items in, but they may be incorrectly overwritten or incorrectly output gated at time of recall. Importantly, incorrect responses generate negative RPEs that punish preceding gating actions, even if some of those actions were appropriate. A preponderance of negative RPEs can thus cause a network to 'give up,' as observed in the no-chunk models when set size exceeds allocated capacity, leading to empty stripes (*Figure 7*). This mechanism is conceptually related to rodent findings in the motor domain, whereby lower DA levels can induce aberrant NoGo learning even for adaptive actions, causing progressive impairments (*Beeler et al., 2012*).

## Chunking can mitigate against giving up via shared credit assignment

The chunk model can combat against using the 'giving-up' strategy: when items are chunked, the chunk stripe is used more frequently and, therefore, has a greater chance of receiving the positive

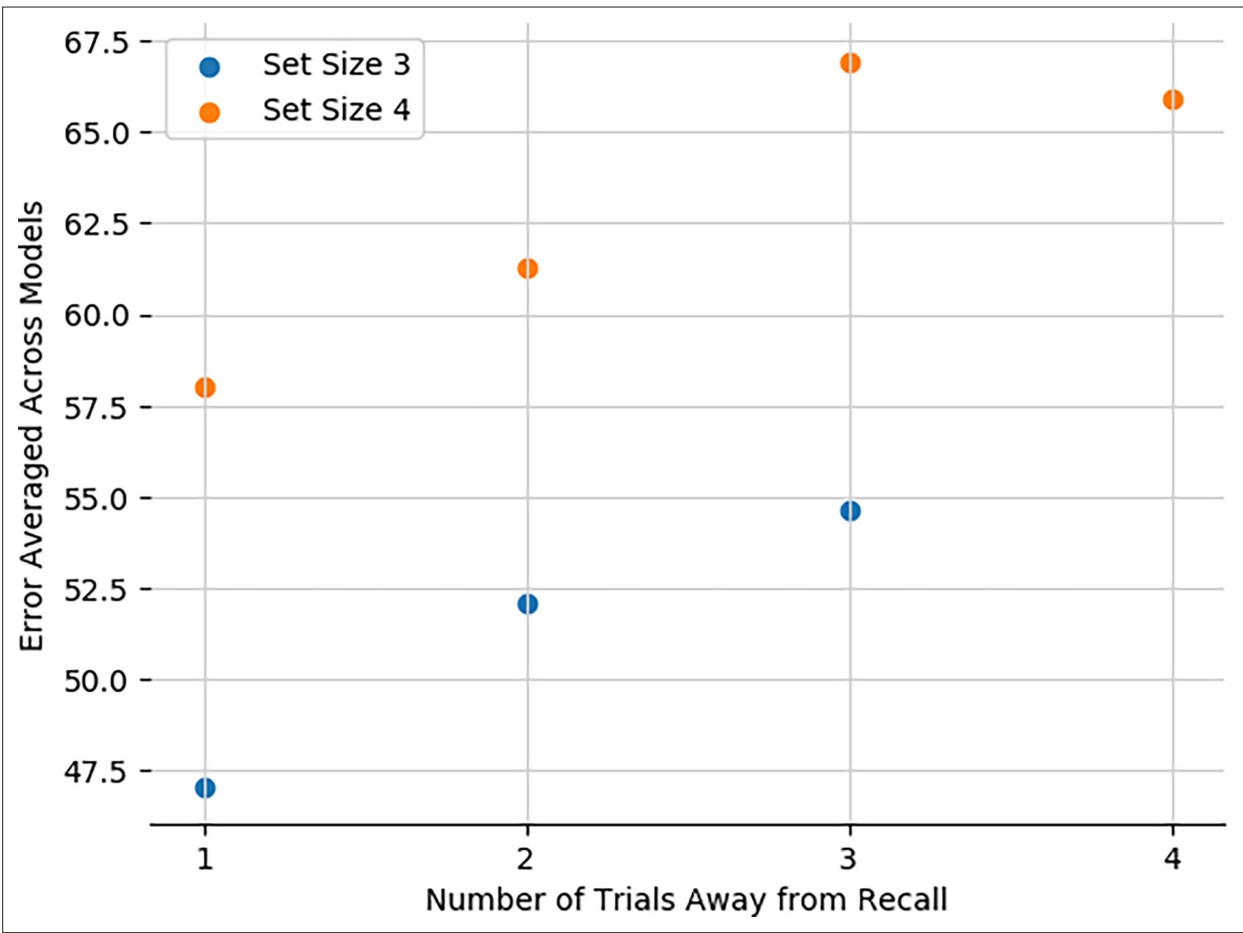

**Figure 11.** Network captures recency effects. Average error on recall trials as a function of the distance in trials between presentation of the relevant stimulus and recall.

reinforcement, and thus benefits from shared credit assignment. The model still has to learn when to use the chunk vs. input stripes, but chunking serves as an aid to the reinforcement learning process. Therefore, the chunk model is also more robust compared to the no-chunk model across various parameter ranges of dopamine. However, the chunk model still fails if the DA dip value is sufficiently larger than the DA burst, for similar 'giving up' reasons (*Figure 10c and e*).

## Network recapitulates human sequential effects in working memory

Finally, we also tested whether the model can reproduce well-documented sequential effects in human working memory studies. Various findings indicate that in WM experiments, humans show higher accuracy for items most recently encountered. Moreover, Oberauer et al. shows how recency effects in humans stem not just from passive decay but specifically from intervening distractors. Our network reproduces these effects, with average error monotonically increasing with number of intervening trials since the relevant stimulus was encountered, in both set sizes 3 and 4 (*Figure 11*). This results from the simple principle whereby with more intervening trials, the gating model is more likely to have updated a corresponding stripe, either replacing it altogether (leading to increased probability of forgetting) or chunking with previous items, thereby increasing average error. The errors are overall smaller in set size 3 because the network is less likely to overwrite an item altogether (it can chunk and still recall one of the items perfectly). Finally, the recency effects asymptotes in set size 4 due to increased opportunities for chunking(indeed, no-chunk networks continued to show increasing errors four trials back; not shown).

## Discussion

Our work synthesizes and reconciles theories of visual WM across multiple levels of abstraction. At the algorithmic level, there has been extensive debate regarding the nature of WM capacity limitations, with experimental results alternately supporting slots or resources theories (*Bays et al., 2009*; *van den Berg et al., 2012*; *Wei et al., 2012*; *Swan and Wyble, 2014*). At the mechanistic level, several studies across species and methods suggest that circuits linking frontal cortex with basal ganglia and thalamus support WM input and output gating (*McNab and Klingberg, 2008*; *Cools et al., 2007*; *Cools et al., 2010*; *Baier et al., 2010*; *Nyberg and Eriksson, 2016*; *Chatham et al., 2014*; *Wilhelm et al., 2023*; *Rikhye et al., 2018*; *Nakajima et al., 2019*). These data accord with the PBWM and related neural network models of PFC-BG gating processes (*Frank et al., 2001*; *O'Reilly and Frank, 2006*; *Hazy et al., 2007*; *Krueger and Dayan, 2009*; *Stocco et al., 2010*; *Badre and Frank, 2012*; *Calderon et al., 2022*). To date, however, these lines of literature have for the most part not intersected. Here, we show that when augmented with continuous population code values and a chunking layer, PBWM comprises a hybrid between slots and resources with resource-like constraints within individual stripes. Moreover, through reinforcement learning, adaptive gating policies can adjust the degree to which behavior mimics primarily slots-like or resource-like as a function task demands. As such, this model accounts for human findings supporting chunking of related items in WM, that such chunking evolves with reward feedback, and is predictive of better performance with increasing task demands across multiple datasets (*Nassar et al., 2018*).

On its surface, PBWM is ostensibly a slot-like model, with distinct PFC clusters ('stripes') corresponding to slots that can be independently gated, giving rise to useful computational properties such as variable binding, indirection, compositionality, and hierarchical generalization (*O'Reilly and Frank, 2006*; *Kriete et al., 2013*; *Collins and Frank, 2013*; *Calderon et al., 2022*). The need for gating also accords with data suggesting that effective WM capacity is related to management of WM content, ie. one's ability to filter out distractors so as to prioritize task-relevant information *Vogel et al., 2005*; *Astle et al., 2014*; *Feldmann-Wüstefeld and Vogel, 2019*, and that such abilities rely on basal ganglia output and function (*McNab and Klingberg, 2008*; *Baier et al., 2010*). However, previous applications of PBWM and related networks have not confronted tasks requiring storage of continuous valued stimuli. In this work we augmented PBWM with a ring attractor layer that resembles that which has previously been applied to such continuous report tasks, supporting population level coding and mergers of nearby attractors (*Wei et al., 2012*; *Edin et al., 2009*; *Nassar et al., 2018*). However, in our network, this layer receives bottom-up input from both the sensory input layer and top-down input from all the PFC stripes, thereby allowing the network to combine sensory information

with the nearest neighbor in memory. Moreover, WM chunking in our model is not obligatory, as the network can learn a gating policy that prioritizes raw sensory inputs (in which case it can represent a given item precisely) or to either replace a currently stored PFC item with the chunked version. As such, the PBWM-chunk model can learn to store more items than the allocated slots capacity by combining representations while incurring the cost of lost precision on the chunked items, giving rise to resource-like behavior. Given this learned policy, the network still may encounter trials where chunking is not possible and all stripes are occupied, leading to guessing and slots-like behavior. Depending on the learned gating policy and the task, the errors look more 'slots-like' or 'resource-like'.

As such, our model addresses key limitations in previous neural models in this domain, in which chunking was obligatory fashion due to perceptual overlap and could not be optimized (***Wei et al., 2012***; ***Nassar et al., 2018***). Instead, PBWM adapts whether or not to chunk as a function of task demands and reward history (***Figure 9***), similar to empirical data (***Nassar et al., 2018***). Furthermore, PBWM can also report only the color of the probed item, unlike previous neural models which were considered accurate as long as the probed color was one of the various populations still active (***Wei et al., 2012***; ***Nassar et al., 2018***).

Critically, to perform adequately, PBWM requires learning appropriate input and output gating policies which are not hard-wired, and indeed involves solving a difficult credit assignment problem (***O'Reilly and Frank, 2006***). At the input gating level, the network must learn whether to gate the chunked or the raw sensory stimulus (via updating of the chunk vs input stripe). Simultaneously it must also learn which stripe to output gate in response to a given probe, which requires coordinating its input and output gating strategies so that they align. The credit assignment problem, understanding which input gating decisions in combination with output gating decisions lead to reward, is difficult. To understand the difficulty, we can look at an example case where the model input gates into stripe 1. However, during read out, since it has not learned the proper binding yet, it gates out from stripe 2, leading to an error and a dopamine dip. In this case, due to an improper output gating decision, both input gating decisions and output gating decisions will be punished. Eventual successful performance requires exploration of alternate gating strategies and reinforcement of those that are effective.

How can chunking help? First, it is important to note that the above problem becomes even more difficult as the number of stripes increases – even if it matches or exceeds the set size (as shown in ***Figure 8***). For example, random exploration and guessing will lead to the correct response (an item being gated into a stripe AND read out from the correct stripe) 50% of the time with two stripes and 33% if the model has three stripes. The general form is: $\frac{(n-1)^{N-1}}{n^N}$ where n is the number of stripes and where N is the set size. For a quick intuition, we assume that the first item is gated into any one of the stripes. We then multiply two probabilities: (1) the probability that the second item is gated anywhere *except* where the first item was stored - which is $n-1/n$ for 1 additional item. This probability is multiplied as many times based on the size minus 1 since the first item is already stored (the power is $N-1$) (2) The probability that the first item is gated out correctly, which is $1/n$. The probability of this correct guess goes down as the number of stripes increases. The difficulty also increases with set size because the network must learn where to input and output gate for each item, and it is also possible for it to overwrite information by updating a stripe. As such, using a smaller number of stripes but allowing for chunking provides a lossy compression strategy that can mitigate this problem and render credit assignment easier, despite the loss of precision. Rather than overwriting information, the network can learn to use the chunk-linked PFC stripe if it is predictive of task success (minimal cost in the reward function for small errors), and moreover, when chunking is 'good enough' the network can leverage repeated reinforcement of the same gating policy to store and read out from the chunked-link PFC stripe, thereby improving credit assignment.

As such, our simulations provide a provocative if somewhat speculative understanding of the nature of WM capacity limitations. It is unlikely that such limitations result from limits in the number of neurons or stripes available in prefrontal cortex, given that discrete estimates on capacity limitations range in the order of three to four items whereas the number of stripes (or equivalent clusters of PFC populations) is orders of magnitude larger (***Frank et al., 2001***). Our simulations show that a limiting step is properly utilizing and managing resources to optimize performance (***Figure 7***), and that it might actually be more effective to limit the number of representations used. Increasing model capacity to four and eight stripes and the resulting comparisons show that the limitation in the model is not simply about number of slots but the complexity of learning. Using WM requires multiple input

and output gating decisions and strategies in tandem with solving and learning a task - this would become trivial with a homunculus dictating what information to store and where to store it. In biology, the PFC has to *learn* these gating strategies: it is not hardwired. This set of experiments helps explain various other WM findings which suggest that effective WM capacity is not just about 'capacity' but rather is also about the ability to filter out irrelevant information, the importance of the task (reward), and experience with the task (experts vs. novice) (*McNab and Klingberg, 2008*; *Astle et al., 2014*; *Nassar et al., 2018*; *Feldmann-Wüstefeld and Vogel, 2019*; *Nakajima et al., 2019*). It also accords with our findings that when exceeding capacity, networks often 'gave up' in the sense that they had more trials in which at least one stripe was empty, due to the influence of negative prediction errors punishing gating policies. As such, we showed that networks require a larger dopamine burst than dip to succeed with increasing task demands. This finding also accords with related data in rodents and our network model in the motor domain, whereby dopamine depletion can cause a progressive 'unlearning' of adaptive strategies (i.e. 'giving up') via aberrant NoGo learning (*Beeler et al., 2012*). This learned Parkinsonism was shown to be related to plasticity in D2 medium spiny neurons (*Beeler et al., 2012*), and this mechanism was recently confirmed to depend on the indirect pathway (*Cheung et al., 2023*).

More generally, our simulations revealed an important role for RL in shaping gating policies as a function of task demands, mimicking normative analysis showing that optimal chunking criterion changes with set size (*Nassar et al., 2018*). In the network, dopamine is a critical component of RL by adjusting synaptic weights into striatal modules that support input and output gating. The need for a healthy balanced dynamic range of DA signals for adaptive performance provides a potential window into a mechanism that can explain deficits in patient populations with altered striatal DA signaling. Whereas much of the literature in patients with schizophrenia and ADHD focuses on limitations in WM capacity, our simulations suggest an alternative whereby altered DA signaling in these populations (*Maia and Frank, 2017*) could influence chunking and efficient use of resources. Our finding that adaptively learned gating policies are important for controlling when and whether to chunk may have implications for recent accounts of patient populations with repetitive negative thinking, which are proposed to arise from failures inherent to learning adaptive mental gating (*Hitchcock and Frank, 2024*). Future work in these patient populations could aim to study these nuances for better understanding of their cognitive deficits.

## Limitations and future directions

There are several limitations to this work. For simplicity, we restricted our simulations to a chunking network with just one chunk-linked PFC stripe and one or more input stripes. In this case, the determining factor for whether stimuli are merged in the chunking layer depends on how close they are in color, lateral inhibition in the chunking layer, and the relative strength of top-down PFC projections to the chunk layer. These parameters were fixed in our simulations and were not formally optimized. A more general model could include multiple chunking layers with a reservoir of effective chunking thresholds (e.g. with varying degrees of top-down influence and lateral inhibition). Depending on the task, the model could learn to chunk more liberally (larger set size - larger threshold) or more restrictively (smaller set size - smaller threshold), by adapting gating policies to rely on PFC stripes linked to these finer or coarser representations. Alternatively, it is possible that a network could learn to adapt these hyperparameters directly within the chunking layer. Furthermore, through development the brain learns the environmental statistics and could learn those threshold parameters on a developmental time scale and could be fine-tuned on a task-by-task basis. Our objective was to explore how far one can get by optimizing only the gating policy via biologically plausible RL rules explored widely in basal ganglia.

Because of its wide application in the literature, we considered tasks in which stimuli can be chunked along a single scalar dimension (color or orientation, both of which have shown evidence for chunking *Nassar et al., 2018*). Future work should explore to what degree these principle could generalize to more complex stimuli where chunking could occur across other more abstract dimensions, depending on the task demands (*Kiyonaga et al., 2017*). This model has the potential to be scaled up and here, we show the core principles for how the chunking gating strategy can be learned via RL.

One key difference is how the task is presented to the model and to humans. Humans are given clear verbal instructions and are able to perform the color wheel task with little to no practice. However, the

model does not receive verbal communication and must learn the task from scratch - random weights. It has no prior experience with how to process the stimuli or how to maintain any stimuli. Humans learn this through development and establish a general gating policy. In a everyday setting, while individuals are not re-learning connections and gating policies to fit individual tasks, they are 'fine-tuning' how they manipulate the information in WM and how to apply their learned policies to adapt to the current task. Experimental results show how reward or task relevance are factors that can tweak gating policies (*O'Reilly and Frank, 2006*; *Nassar et al., 2018*).

## Acknowledgements

Aneri Soni was supported by NIMH training grant T32MH115895 (PI's: Frank, Badre, Moore). The project was supported by ONR MURI Award N00014-23-1-2792 and NIMH R01 MH084840-08A1. Computing hardware was supported by NIH Office of the Director grant S10OD025181.

## Additional information

### Competing interests

Michael J Frank: Reviewing editor, eLife. The other author declares that no competing interests exist.

### Funding

| Funder | Grant reference number | Author |
|---|---|---|
| National Institute of Mental Health | T32MH115895 | Aneri Soni |
| Office of Naval Research | N00014-23-1-2792 | Aneri Soni<br>Michael J Frank |
| NIH Office of the Director | S10OD025181 | Aneri Soni<br>Michael J Frank |
| National Institute of Mental Health | R01 MH084840-08A1 | Michael J Frank |

The funders had no role in study design, data collection and interpretation, or the decision to submit the work for publication.

### Author contributions

Aneri Soni, Conceptualization, Resources, Data curation, Software, Formal analysis, Validation, Investigation, Visualization, Methodology, Writing – original draft, Project administration, Writing – review and editing; Michael J Frank, Conceptualization, Resources, Software, Formal analysis, Supervision, Funding acquisition, Methodology, Project administration, Writing – review and editing

### Author ORCIDs

Aneri Soni https://orcid.org/0000-0001-6410-8414
Michael J Frank https://orcid.org/0000-0001-8451-0523

Reviewer #1 (Public review): https://doi.org/10.7554/eLife.97894.3.sa1
Reviewer #2 (Public review): https://doi.org/10.7554/eLife.97894.3.sa2
Author response https://doi.org/10.7554/eLife.97894.3.sa3

## Additional files

### Supplementary files

MDAR checklist

## Data availability

The current manuscript is a computational study. Data generation is part of the modeling code that is located on Github (https://github.com/AneriSoni/workingmemory, copy archived at *Soni, 2024*) and made available in the manuscript.

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

# Appendix 1

## Neural model simulations

For *each frontal stripe*, the corresponding striatal gating layers consisted of 24 distributed units (12 Go and 12 NoGo) which learn the probability of obtaining a reward if the stimulus in question is gated into, or out of, it's respective working memory stripe. The number of units is proportional to the number of stripes and, therefore, in the model iterations with four or eight stripes, the striatal gating layers are larger. In each module, a Gpi/Thal (globus pallidas internus/thalamus) unit implements a gating signal and is activated when relatively more striatal Go than NoGo units are active (subject to inhibitory competition from other GPi/Thal units that modulate gating of neighboring stripes *O'Reilly and Frank, 2006*). Thus the GPi/Thal units summarize the contributions of multiple interacting layers that implement gating among the globus pallidus, subthalamic nucleus, and thalamus as simulated in more detailed networks of a single BG circuit (*Frank and Claus, 2006*), in these larger-scale networks we abstract away from these details. For input-gating circuits, GPi/Thal activation induces maintenance of activation states in the corresponding frontal maintenance layer (*Frank et al., 2001*; *O'Reilly and Frank, 2006*). For output-gating circuits, the GPi/Thal activation results in information flow from the frontal maintenance layer to the frontal output layer. This output layer projects to the decision circuit, such that only output-gated representations influence response selection.

The task in this experiment was the color visual working memory task. Each trial consisted of stimulus presentation, during which stimuli could be gated into corresponding PFC areas, followed by another phase in which all input stimuli were removed and the network had to rely on maintained PFC representations in order to respond. The frontal stripes for each of the stimulus dimensions could independently maintain representations of these stimulus dimensions in $PFC_{MaintDeep}$, subject to gating signals from the BG. Initially, a 'Go bias' encourages exploratory updating (and subsequent maintenance) due to novelty; these gating signals are then reinforced to the extent that the frontal representations come to be predictive of reward *O'Reilly and Frank, 2006*. However, not all maintained $PFC_{MaintDeep}$ representations influence decision in the response circuitry, only those that are also represented in $PFC_{Out}$ due to output gating signals. Thus in a given trial, color of the current stimulus may be represented in $PFC_{MaintDeep}$, but the output gating will dictate the ultimate model response.

## Neural model implementational details

The model is implemented using the Leabra framework (*O'Reilly and Munakata, 2000*; *O'Reilly and Munakata, 2019*), with the new version of the emergent neural simulation software, which contains extensive documentation and examples that can be run in Python or the Go language (https://github.com/emer/emergent). All of the computational models, and the code to perform the analysis, are available and will be published on our GitHub account. The PBWM network used here simulates the anatomical projections and physiological properties of the BG circuitry in learning, working memory and decision making (*Frank, 2005*; *O'Reilly and Frank, 2006*). Leabra uses point neurons with excitatory, inhibitory, and leak conductances contributing to an integrated membrane potential, which is then thresholded and transformed to produce a rate code output communicated to other units. Dopamine in the BG modifies activity in Go and NoGo units in the striatum, and this modulation of activity affects both the propensity for overall gating (Go relative to NoGo activity), and activity-dependent plasticity that occurs during reward prediction errors (*Frank, 2005*; *Wiecki et al., 2009*; *Beeler et al., 2012*; *Jaskir and Frank, 2023*). Both of these functions are detailed below.

The membrane potential $V_m$ is updated as a function of ionic conductances $g$ with reversal (driving) potentials $E$ according to the following differential equation:

$$C_m \frac{dV_m}{dt} = \begin{aligned} &g_e(t)\bar{g}_e(E_e - V_m) + \\ &g_i(t)\bar{g}_i(E_i - V_m) + \\ &g_l(t)\bar{g}_l(E_l - V_m) + ..., \end{aligned} \tag{8}$$

where $C_m$ is the membrane capacitance and determines the time constant with which the voltage can change, and subscripts $e$, $l$ and $i$ refer to excitatory, leak, and inhibitory channels, respectively (and '…' refers to the possibility of adding other channels implementing neural accommodation and hysteresis). Following electrophysiological convention, the overall conductance for each channel $c$

is decomposed into a time-varying component $g_c(t)$ computed as a function of the dynamic state of the network, and a constant $\overline{g_c}$ that controls the relative influence of the different conductances. The equilibrium potential can be written in a simplified form by setting the excitatory driving potential ($E_e$) to 1 and the leak and inhibitory driving potentials ($E_l$ and $E_i$) of 0:

$$V_m^\infty = \frac{g_e \overline{g_e}}{g_e \overline{g_e} + g_l \overline{g_l} + g_i \overline{g_i}} \tag{9}$$

which shows that the neuron is computing a balance between excitation and the opposing forces of leak and inhibition. The excitatory net input/conductance $g_e(t)$ is computed as the proportion of open excitatory channels as a function of sending activations times the weight values:

$$g_e(t) = \langle x_i * wi \rangle = \frac{1}{n} \sum_i x_i w_i \tag{10}$$

The inhibitory conductance is computed as described in the next section, and leak is a constant.

Activation communicated to other cells ($y_j$) is a thresholded ($\Theta$) sigmoidal function of the membrane potential with gain parameter $\gamma$:

$$y = \frac{1}{\left(1 + \frac{1}{\gamma[g_e - g_e^\Theta]_+}\right)} \tag{11}$$

where $g_e^\Theta$ is the level of excitatory input conductance that would put the equilibrium membrane potential right at the firing threshold $\Theta$ and depends on the level of inhibition and leak.

$$g_e^\Theta = \frac{g_i(E_i - \Theta) + g_l(E_l - \Theta)}{\Theta - E_e} \tag{12}$$

## Inhibition within layers

For within layer lateral inhibition, Leabra uses feedforward and feedback (FFFB) inhibition, allowing the $gi$ values for each neuron to be adjusted as a function of total net input to a layer (feedforward inhibition) as well as the total excitatory activity of that layer (feedback inhibition). The average net input (excitatory conductance) to a layer is just the average of the of each unit indexed by in the layer:

$$\langle g_e \rangle = \sum_n \frac{1}{n} ge_i$$

Similarly, the average activation is just the average of the activation values ($y_i$):

$$\langle y \rangle = \sum_n \frac{1}{n} y_i$$

The overall inhibitory conductance is just the sum of the two terms (ff and fb), with an overall inhibitory gain constant factor Gi:

$$g_i(t) = \text{Gi} \left[ \text{ff}(t) + \text{fb}(t) \right]$$

This Gi factor is typically the only parameter manipulated to determine overall layer activity level. The default value is 1.8 (but is reduced in the chunk layer, see below).

The feedforward (ff) term is:

$$\text{ff}(t) = \text{FF} \left[ \langle g_e \rangle - \text{FF0} \right]_+$$

where FF is a constant gain factor for the feedforward component (set to 1.0 by default), and FF0 is a constant offset (set to 0.1 by default).

The feedback (fb) term is:

$$\text{fb}(t) = \text{fb}(t-1) + dt \left[ \text{FB} \langle y \rangle - \text{fb}(t-1) \right]$$

where FB is the overall gain factor for the feedback component (0.5 default), dt is the time constant for integrating the feedback inhibition (0.7 default), and the t-1 indicates the previous value of the feedback inhibition.

## Connectivity

The connectivity of the BG network is critical, and is thus summarized here (see *Frank and Claus, 2006* and *O'Reilly and Frank, 2006* for details and references). Unless stated otherwise, projections are fully connected (that is all units from the source region target the destination region, with a randomly initialized synaptic weight matrix). However, the units in PFC, Striatum, GPiThal are all organized with columnar structure. Units in the first stripe of PFC represent one set of representations and project to a single column of Go and NoGo units in the Striatum, which in turn project to the corresponding column in GPiThal. Each Thalamic unit is reciprocally connected with the associated column in PFC. This connectivity is similar to that described by anatomical studies, in which the same cortical region that projects to the striatum is modulated by the output through the BG circuitry and Thalamus.

Dopamine units in the SNc project to the entire Striatum, but with different projections to encode the effects of D1 receptors in Go neurons and D2 receptors in NoGo neurons. With increased dopamine, Go units are excited while NoGo units are inhibited, and vice-versa with lowered dopamine levels. The particular set of units that are impacted by dopamine is determined by those receiving excitatory input from sensory cortex and PFC. Thus dopamine modulates this activity, thereby affecting the relative balance of Go vs NoGo activity in those units activated by cortex. This impact of dopamine on Go/NoGo activity levels influences both the propensity for gating (during response selection) and learning, as described next.

## Learning

Learning in the model is activity-dependent and uses a biologically motivated homeostatic learning rule called the eXtended Contrastive Attractor Learning (XCAL) rule. The empirical learning function (called the XCAL dWt function) approximates that observed from a highly biologically detailed computational model of the known synaptic plasticity mechanisms, by *Urakubo et al., 2008*; see *O'Reilly et al., 2024*. XCAL uses a simple piecewise-linear function, described below, that emerges from it. This XCAL dWt function resembles the BCM learning function, where weight changes are a function of presynaptic activation $x$ and postsynaptic activation $y$ relative to a floating threshold (approximating effects of calcium levels), and is functionally similar to contrastive Hebbian learning. The floating threshold determines the amount of activity needed to elicit LTP vs LTD.

$$\Delta w = f_{xcal}\left(xy, \theta_p\right)$$

$$f_{xcal}(xy, \theta_p) = \begin{cases} (xy - \theta_p) & \text{if } xy > \theta_p\theta_d \\ -xy(1 - \theta_d)/\theta_d, & \text{otherwise} \end{cases}$$

where $\theta_p$ is the floating threshold and $\theta_d = 0.1$ is a constant that determines the point where the function reverses direction (i.e., back toward zero within the weight decrease regime).

In XCAL, error-driven learning is accommodated by allowing the floating threshold to vary as a function of recent activity (*O'Reilly et al., 2024*). Weights are increased if activity states during the outcome are greater than their recent levels (i.e. activity states while the network is generating a response), and conversely, weights decrease if the activity levels go down relative to prior states. Thus, we can think of the recent activity levels (the threshold) as reflecting expectations which are subsequently compared to actual outcomes, with the difference (or 'error') driving learning. Thus XCAL is closely related to contrastive Hebbian learning, where weight changes are determined by changes in activation. As we will see below, in PBWM and BG models, the error in gating signals is driven by changes in activation resulting from dopaminergic RPEs (*Frank, 2005*; *O'Reilly and Frank, 2006*; *Frank and Badre, 2012*).

## Striatal learning function

Synaptic connection weights in striatal units were trained using a reinforcement learning version of Leabra. The learning algorithm involves two phases, and is more biologically plausible than standard error backpropagation. In the *minus phase*, the network settles into activity states based on input stimuli and its synaptic weights, ultimately 'choosing' a response. In the *plus phase*, the network

resettles in the same manner, with the only difference being a change in simulated dopamine: an increase of SNc unit firing for positive reward prediction errors, and a decrease for negative prediction errors (*Frank, 2005*; *O'Reilly and Frank, 2006*). This change in dopamine modifies Go and NoGo activity levels, and because synaptic strengths are adjusted as a function of activity levels relative to the floating threshold (previous activity levels prior to the RPE), this functionality also influences what is learned.

For the large-scale BG-PFC models used here and in *O'Reilly and Frank, 2006* some abstractions are used. Each stripe (group of units) in the Striatum layer is divided into Go vs. NoGo in an alternating fashion. The DA input from the SNc modulates these unit activations in the update phase by providing extra excitatory current to Go and extra inhibitory current to the NoGo units in proportion to the positive magnitude of the DA signal, and vice-versa for negative DA magnitude. This reflects the opposing influences of DA on these neurons (*Frank, 2005*; *Gerfen, 2000*; *Shen et al., 2008*). The update phase DA signal reflects the critic system's reward prediction error (RPE) produced by gating signals (see below) – that is, if the PFC state is predictive of reward, the striatal units will be reinforced. Learning on weights into the Go/NoGo units is based on the activation delta that results from this RPE using the same XCAL dWt function defined above (which is functionally similar to contrastive Hebbian learning).

## Dopamine and prediction errors in the 'critic'

We used a simplified version of the critic in these simulations because they do not depend on the differences between different algorithms (e.g. temporal difference learning or 'PVLV,' the algorithm used in our other BG-PFC networks *O'Reilly and Frank, 2006*; *Hazy et al., 2007*). The algorithm we used for generating dopaminergic prediction errors corresponds to a basic Rescorla-Wagner delta rule, as also reported in *Frank and Badre, 2012*. The use of a simple delta rule allowed us to confirm that simulation results do not depend on the details of the basic RL algorithm. The reward prediction error (RPE or δ) is the difference between the delivered reward (R) and the expected reward (V). The expected reward is calculated by a RewardPrediction unit which calculates the value of the gating actions based on the previously earned rewards.

$$\delta = R - V \tag{13}$$

An updated V in this RewardPrediction unit is calculated after the RPE is determined, with learning rate α:

$$V_{updated} = V + \alpha(\delta) \tag{14}$$

## GPiThal units

The GPiThal units provide a simplified version of the GPi/Thalamus layers abstracted from the full circuitry implemented in more basic versions of the BG circuit (*Frank and Claus, 2006*, e.g.). They receive a net input that reflects the normalized Go - NoGo activations in the corresponding Striatum stripe:

$$\alpha_j = \left[ \frac{\sum Go - \sum NoGo}{\sum Go + \sum NoGo} \right]_+ \tag{15}$$

(where []₊ indicates that only the positive part is taken; when there is more NoGo than Go, the net input is 0). This net input then drives standard Leabra point neuron activation dynamics, with FFFB inhibitory competition dynamics that cause stripes to compete for both input and output gating of PFC. This dynamic is consistent with the notion that competition/selection takes place primarily in the smaller GP/GPi areas, and not much in the much larger striatum e.g. (*Jaeger et al., 1994*). The resulting GPiThal activation then provides the gating update signal to the PFC: if the corresponding GPiThal unit is active (above a minimum threshold; .1), then active maintenance currents in the PFC are toggled.

## PFC maintenance

PFC active maintenance is supported in part by excitatory ionic conductances that are toggled by Go firing from the GPiThal layers. This is implemented with an extra excitatory ion channel in the basic $V_m$ update *Equation 9*. This channel has a conductance value of 0.5 when active. See (*Frank et al., 2001*) for further discussion of this kind of maintenance mechanism, which has been proposed

by several researchers (e.g.) (*Lisman et al., 1998*; *Dilmore et al., 1999*; *Gorelova and Yang, 2000*; *Durstewitz et al., 2000*). The first opportunity to toggle PFC maintenance occurs at the end of the first plus phase, and then again at the end of the second plus phase (third phase of settling). Thus, a complete update can be triggered by two Go's in a row, and it is almost always the case that if a Go fires the first time, it will fire the next, because Striatum firing is primarily driven by sensory inputs, which remain constant.

## Continuous stimuli

Previous applications of PBWM consisted of discrete stimuli. For this project, the stimuli were made continuous on a ring from 0 to 360 to mimic the color wheel. The Gaussian bump width is 0.15 and there are 20 units per layer.

Stimuli are presented sequentially. During a store trial, the color and orientation are presented simultaneously and remain active for the duration of the gating decision and until the next trial begins. Each trial is 100 ms long (10 Hz alpha frequency) and is organized into four quarters. Each quarter lasts 25 ms (40 Hz, gamma frequency). The first three quarters form the expectation (minus phase) and the last quarter is the outcome (plus phase). The difference between the minus and plus phase dictates learning. For further details on the cycles (updating of each neurons membrane potential) and the timing of each trial, see O'Reilly et al.

## Layer sizes and inner mechanics

### Input, chunk, and output layers each have 20 neurons

The PFC is composed of four layers: maintenance and output layers, each with their superficial and deep components. When sensory inputs are presented, the superficial maintenance layers always represent them transiently. Only when gated, inputs are then maintained (due to thalamocortical innervation of deep layers) over time in the absence of subsequent input. The output layers manage output gating in an analogous fashion. All information stored in WM (maintenance deep layer) is presented to the output superficial layer. Only when the corresponding basal ganglia thalamic circuit issues an output gating signal will this information propagate to the output deep layer, and thus be accessible for 'read out' by the hidden and output layers of the network.

Each PFC layer is an array of neurons that has 20 units for each stripe. For most of the experiments here, we used two stripes. Each stripe has 20 neurons to match with input/output layers. (This model could expanded where the representations in PFC would be distributed, as in some other applications of PBWM).

The control input layer (which represents the orientation and the store/recall instruction) has size based on the number of orientations. For four discrete orientations, the size of this layer is 4+1 (for ignore stimulus)+1 (for recall)=6 units. The ignore stimulus is not discussed in this paper, but allow for distractors to be presented, where the model would have to learn to ignore (not update) those stimuli. Early qualitative results suggest that findings hold even when the model is presented with stimuli that it must ignore. During a recall trial, the model input is activity in this recall unit in the control input layer, along with the orientation that should be recalled. No color is presented at this time.

The Striatum is compose of Matrix Go (D1-containing) and Matrix NoGo (D2-containing) units. Each matrix layer is further broken down into two sub-layers which represent input and output gating decisions. Each of these sublayers is an array of units of size number of stripes x size of control input layer. For four orientations and two stripes, each sublayer has 2×6 (see previous paragraph)=12 units. This means for both input and output gating, each of the Matrix Go and NoGo layers have 24 units. The matrix learns to perform a gating operation to the corresponding PFC stripe based on the given control input (e.g. store orientation 1; recall orientation 3, etc).

The globus pallidus externus (GPe) layer has one unit for each input and output stripe; for two stripes, the size of this layer is 4. These units receive inputs from corresponding Matrix NoGo layers, preventing gating of the corresponding input and output stripes when the NoGo units are active (i.e. when they have learned that these gating operations are not useful).

The globus pallidus internus (Gpi)/Thalamus (dorsomedial thalamus) layer receives projections from the Matrix Go layer to induce gating, but competing inputs from the GPe to prevent gating. The relative balance of Go vs NoGo signals for a given stripe thus determine, along with inhibitory competition within this layer, whether a given PFC stripe is gated. This implementation abstracts

over the details of thalamic representations, combining GPi and Thalamus into one functional layer *O'Reilly and Frank, 2006* but see more detailed implementations of this functionality *Frank and Claus, 2006*; *Collins and Frank, 2013*.

The reinforcement learning component is broken into three single unit layers, as described above: the Reward (which indicates the reward the model receives), the RewPred (which is the model's predicted reward), and the SNc (substantia nigra pars compacta) dopaminergic cells which calculates the difference between the two and generates the reward prediction error (RPE).

## Hyperparameter search

The PBWM is a well-established framework with many parameters. To maintain consistency with prior work and avoid an overly complex parameter search, default parameters were used with the following exceptions. A hyperparameter search was optimized for baseline model performance for parameters most relevant to this project: chunking layer inhibition (1.4), chunking layer gain (5), relative weight scales (connectivity strength) between the chunk layer and PFC layer (0.8), input layer to the chunk layer (0.7), and PFC layer to chunk layer (0.2; this lower value ensures that the chunk layer primarily reflects the input and is only influence by PFC if the nearest neighbor overlaps with the input). The striatal learning rate was set to default of 0.04 for set size 2 and 4, but changed to 0.06 for set size 3, which improved performance for the OCV analysis in that set size (but overall performance is not substantially altered). The relative impact of striatal NoGo vs Go activity on GpiThal layer for inducing a gating signal was set to 1.25. Finally to accommodate continuous representations with large Gaussian bump widths in PFC we increased the PFC gain parameter to 5 (otherwise the PFC would not maintain units with lesser activation on the tails of the continuous distribution). The reward function was also altered due to the continuous nature of the outputs, such that rewards are determined based on decoded values of the output layer across the population code relative to the true color that should have been reported, as a continuous linear function of the error. We also varied dopamine burst and dip gain across a wide range to explore its impact as described in the main text.

## Neural model output analysis

### Out of cluster variance (OCV) analysis

The OCV analysis was performed using set size of 3, largely for interpretability. (This analysis becomes convoluted when increasing the set size: the combinations of possible chunking make it hard to divide the stimuli into 'other stimuli' and 'probed stimuli').

The OCV analysis was performed on a subset of trials to specifically test whether the network can leverage chunking abilities to free up resources for other items outside of the chunk. In these trials the 'chunkable' items were presented first and were within 20 degrees away from each other (in color space). The third item is at least 50 degrees away from one of the stimuli. The same procedure was applied to the no-chunk control. If the network chunks the first two items it should then be more likely to store and recall the third item.

The OCV of a single trial is the variance (across the color dimension) of all stimuli in the trial except for the probed item (*Nassar et al., 2018*). The OCV is calculated for each trial and plotted against error on that trial. Since the stimuli and trials are randomly generated, some parts of the graph will be more densely populated. To combat this issue, the trials were binned so that each bin has an equal number of trials and the graph has 20 bins.

