## [Editor Report · eLife Assessment]

This **important** work proposes a neural network model of interactions between the prefrontal cortex and basal ganglia to implement adaptive resource allocation in working memory, where the gating strategies for storage are adjusted by reinforcement learning. Numerical simulations provide **convincing** evidence for the superiority of the model in improving effective capacity, optimizing resource management, and reducing error rates, as well as for its human-like performance. This work will be of broad interest to computational and cognitive neuroscientists, and may also interest machine-learning researchers who seek to develop brain-inspired machine-learning algorithms for memory.

---

## [Referee Report · Reviewer #1 (Public review)]

Summary:

In this research, Soni and Frank investigate the network mechanisms underlying capacity limitations in working memory from a new perspective, with a focus on Visual Working Memory (VWM). The authors have advanced beyond the classical neural network model, which incorporates the prefrontal cortex and basal ganglia (PBWM), by introducing an adaptive chunking variant. This model is trained using a biologically-plausible, dopaminergic reinforcement learning framework. The adaptive chunking mechanism is particularly well-suited to the VWM tasks involving continuous stimuli and elegantly integrates the 'slot' and 'resource' theories of working memory constraints. The chunk-augmented PBWM operates as a slot-like system with resource-like limitations.

Through numerical simulations under various conditions, Soni and Frank demonstrate the performance of the chunk-augmented PBWM model surpass the no-chunk control model. The improvements are evident in enhanced effective capacity, optimized resource management, and reduced error rates. The retention of these benefits, even with increased capacity allocation, suggests that working memory limitations are due to a combination of factors, including the efficient credit assignment that are learned flexibly through reinforcement learning. In essence, this work addresses fundamental questions related to a computational working memory limitation using a biologically-inspired neural network, thus has implications for conditions such as Parkinson's disease, ADHD and schizophrenia.

Strengths:

The integration of mechanistic flexibility, reconciling two theories for WM capacity into a single unified model, results in a neural network that is both more adaptive and human-like. Building on the PBWM framework ensures the robustness of the findings. The addition of the chunking mechanism tailors the original model for continuous visual stimuli. Chunk-stripe mechanisms contribute to the 'resource' aspect, while input-stripes contribute to the 'slot' aspect. This combined network architecture enables flexible and diverse computational functions, enhancing performance beyond that of the classical model.

Moreover, unlike previous studies that design networks for specific task demands, the proposed network model can dynamically adapt to varying task demands by optimizing the chunking gating policy through RL.

The implementation of a dopaminergic reinforcement learning protocol, as opposed to a hard-wired design, leads to the emergence of strategic gating mechanisms that enhance the network's computational flexibility and adaptability. These gating strategies are vital for VWM tasks and are developed in a manner consistent with ecological and evolutionary learning held by human. Further examination of how reward prediction error signals, both positive and negative, collaborate to refine gating strategies reveals the crucial role of reward feedback in fine-tuning the working memory computations and the model's behavior, aligning with the current neuroscientific understanding that reward matters.

Assessing the impact of a healthy balance of dopaminergic RPE signals on information manipulation holds implications for patients with altered striatal dopaminergic signaling.

Comments on revisions:

In the revised version, the authors have thoroughly addressed all the questions raised in my previous review. They have clarified the model architecture, provided detailed explanations of the training process, and elaborated on the convergence of the optimization.

Additionally, Reviewer 2 made a very constructive suggestion: Can related cognitive functions or phenomena emerge from the model? The newly added analysis and results highlighting the recency effect directly address this question and significantly strengthen the paper.

---

## [Referee Report · Reviewer #2 (Public review)]

Summary:

This paper utilizes a neural network model to investigate how the brain employs an adaptive chunking strategy to effectively enhance working memory capacity, which is a classical and significant question in cognitive neuroscience. By integrating perspectives from both the 'slot model' and 'limited resource models,' the authors adopted a neural network model encompassing the prefrontal cortex and basal ganglia, introduced an adaptive chunking strategy, and proposed a novel hybrid model. The study demonstrates that the brain can adaptively bind various visual stimuli into a single chunk based on the similarity of color features (a continuous variable) among items in visual working memory, thereby improving working memory efficiency. Additionally, it suggests that the limited capacity of working memory arises from the computational characteristics of the neural system, rather than anatomical constraints.

Strengths:

The neural network model utilized in this paper effectively integrates perspectives from both slot models and resource models (i.e., resource-like constraints within a slot-like system). This methodological innovation provides a better explanation for the limited capacity of working memory. By simulating the neural networks of the prefrontal cortex and basal ganglia, the model demonstrates how to optimize working memory storage and retrieval strategies through reinforcement learning (i.e., the efficient management of access to and from working memory). This biological simulation offers a novel perspective on human working memory and provides new explanations for the working memory difficulties observed in patients with Parkinson's disease and other disorders. Furthermore, the effectiveness of the model has been validated through computational simulation experiments, yielding reliable and robust predictions.

Comments on revisions:

The authors have already answered all my questions.

---

## [Author Response]

The following is the authors’ response to the original reviews.

**eLife Assessment**
This important work proposes a neural network model of interactions between the prefrontal cortex and basal ganglia to implement adaptive resource allocation in working memory, where the gating strategies for storage are adjusted by reinforcement learning. Numerical simulations provide convincing evidence for the superiority of the model in improving effective capacity, optimizing resource management, and reducing error rates, as well as solid evidence for its human-like performance. The paper could be strengthened further by a more thorough comparison of model predictions with human behavior and by improved clarity in presentation. This work will be of broad interest to computational and cognitive neuroscientists, and may also interest machine-learning researchers who seek to develop brain-inspired machine-learning algorithms for memory.

We thank the reviewers for their thorough and constructive comments, which have helped us clarify, augment and solidify our work. Regarding the suggestion to include a “more thorough comparison with with human behavior”, we believe this comment reflects one of the reviewer’s suggestion to compare with sequential order effects. We now include a new section with simulations showing that the network exhibits clear recency effects in accordance with the literature, and where such recency effects are known to be related to WM interference and not due to passive decay. Overall our work makes substantial contact with human behavioral patterns that have been documented in the human literature (and which as far as we know have not been jointly captured by any one model), such as the shape of the error distributions, including probability of recall and variable precision; attraction to recently presented items, sensitivity to reinforcement history, set-size dependent chunking, recency effects, dopamine manipulation effects, as well of a range of human data linking capacity limitations to frontostriatal function. It also provides a theoretical proposal for the well established phenomenon of capacity limitations in humans, suggesting that they arise due to difficulty in WM management.

Below we address each reviewer individually, responding to each comment and providing the relevant location in the paper that the changes and additions were made. Reviewer responses are included in blue/bold for clarity.

**Public Reviews:**
**Reviewer 1**:

Thank you for your comments. We appreciate your statements of the strengths of this paper and your suggestions to improve this paper.

First, the method section appears somewhat challenging to follow. To enhance clarity, it might be beneficial to include a figure illustrating the overall model architecture. This visual aid could provide readers with a clearer understanding of the overall network model.Additionally, the structure depicted in Figure 2 could be potentially confusing. Notably, the absence of an arrow pointing from the thalamus to the PFC and the apparent presence of two separate pathways, one from sensory input to the PFC and another from sensory input to the BG and then to the thalamus, may lead to confusion. While I recognize that Figure 2 aims to explain network gating, there is room for improvement in presenting the content accurately.

As suggested, we added a figure (new figure 2) illustrating the overall model architecture before expanding it to show the chunking circuitry. This figure also shows the projections from thalamus to PFC (we preserve the previous figure 2, now figure 3, as an example sequence of network gating decisions, in more abstract form to help facilitate a functional understanding of the sequence of events without too much clutter). We also made several other general clarifications to the methods sections to make it more transparent and easier to follow, as per your suggestions.

Still, for the method part, it would enhance clarity to explicitly differentiate between predesigned (fixed) components and trainable components. Specifically, does the supplementary material state that synaptic connection weights in striatal units (Go&NoGo) are trained using XCAL, while other components, such as those in the PFC and lateral inhibition, are not trained (I found some sentences in 'Limitations and Future Directions')?

We have now explicitly specified learned and fixed components. We have further explained the role of XCAL and how striatal Go/NoGo weights are trained. We have also added clarification on how gating policies are learned via eligibility traces and synaptic tags.

I'm not sure about the training process shown in Figure 8. It appears that the training may not have been completed, given that the blue line representing the chunk stripe is still ascending at the endpoint. The weights depicted in panel (d) seem to correspond with those shown in panels (b) and (c), no? Then, how is the optimization process determined to be finished? Alternatively, could it be stated that these weight differences approach a certain value asymptotically? It would be better to clarify the convergence criteria of the optimization process.

The training process has been clarified and we specify (in the last paragraph of the Base PBWM Model) how we determine when training is complete. We also can confirm that the network behavior has stabilized in learning even if the Go/NoGo weights continue to grow over time for the chunked layer (due to imperfect performance and reinforcement of the chunk gating strategy).

**Reviewer 2:**

Thank you for your comments. We appreciate your notes on the strengths of the paper and your suggestions to help improve the paper.

The model employs a spiking neural network, which is relatively complex. Additionally, while this paper validates the effectiveness of chunking strategies used by the brain to enhance working memory efficiency through computational simulations, further comparison with related phenomena observed in cognitive neuroscience experiments on limited working memory capacity, such as the recency effect, is necessary to verify its generalizability.

Thank you for proposing we add in more connections with human WM. Based on your specific recommendation, we have included the section “Network recapitulates human sequential effects in working memory.” where we discuss recency effects in human working memory and how our model recapitulates this effect. We have also made the connections to human data and human work more explicit throughout the manuscript (Figure 4c). As noted in response to the assessment, we believe our model does make contact with a wide variety of cognitive neuroscience data in human WM, such as the shape of the error distributions, including probability of recall and variable precision; attraction to recently presented items, sensitivity to

reinforcement history, set-size dependent chunking, recency effects, and dopamine manipulation effects, as well of a range of human data linking capacity limitations to frontostriatal function. It also provides a theoretical proposal for the well established phenomenon of capacity limitations in humans, suggesting that they arise due to difficulty in WM management.

**Recommendations For The Authors:**
**Reviewer 1**:I appreciate the authors' clear discussion of the limitations of this work in the section "Limitations and Future Directions". The development of a comprehensive model framework to overcome these constraints should require a separate paper, though, I am curious if the authors have attempted any experiments, such as using two identically designed chunking layers, that could partially support the assumptions presented in the paper.

Expanding the number of chunking layers is a great future direction. We felt that it was most effective for this paper to begin with a minimal set up with proof of concept. We hypothesize that, given our results, a reinforcement learning algorithm would be able to learn to select the best level of abstraction (degree of chunking) in more continuous form, but would require more experience across a range of tasks to do so.

I'm not sure whether it's appropriate that "Frontostriatal Chunking Gating..." precedes "Dopamine Balance is...", maybe it would be better to reverse the order thus avoiding the need to mention the role of dopamine before delving into the details. Additionally, including a summary at the end of the Introduction, outlining how the paper is organized, could provide readers with a clear roadmap of the forthcoming content.

We appreciate this suggestion. After careful thought, we wanted to preserve the order because we felt it was important to make the direct connection between set size and stripe usage following the discussion on performance based on increasing stripes.

The authors could improve the overall polish of the paper. The equations in the Method section are somewhat confusing: Eq. (2) appears incorrect, as it lacks a weight w_i and n should presumably be in the denominator. For Eq. (3), the comma should be replaced with'... It would be advisable to cross-reference these equations with the original O'Reilly and Frank paper for consistency.

Thank you for pointing out the errors in the method equations- those equations were indeed rendering incorrectly. We have fixed this problem.

Additionally, there are frequent instances of missing figure and reference citations (many '?'s), and it would be beneficial to maintain consistent citation formatting throughout the paper: sometimes citations are presented as "key/query coding (Traylor, Merullo, Frank, and Pavlick, 2024; see also Swan and Wyble, 2014)", while other times they are written as "function (O'Reilly & Frank, 2006)"...Lastly, there is an empty '3.1' section in the supplementary material that should be addressed.

The citation issues were fixed. The supplementary information was cleaned and the missing section was removed. Thank you for mentioning these errors.

**Reviewer 2:**

Thank you for the following recommendations and suggestions. We respond to each individual point based on the numbering system used in your review.

(1) This paper utilizes the experimental paradigm of visual working memory, in which different visual stimuli are sequentially loaded into the working memory system, and the accuracy of memory for these stimuli is calculated.The authors could further plot the memory accuracy curve as the number of items (N) increases, under both chunking and non-chunking strategies. This would allow for the examination of whether memory accuracy suddenly declines at a specific value of N (denoted as Nc), thereby determining the limited capacity of working memory within this experimental framework, which is about 4 different items or chunks. Additionally, it could be investigated whether the value of Nc is larger when the chunking strategy is applied.

We have included an additional plot (Probability of Recall) as a supplemental figure to Figure 5 to explore the probability of recall as a function of set size for both chunking and no chunking models. This plot shows that the chunking model increases probability of recall when set size exceeds allocated capacity (but that nevertheless both models show decreases in recall with set size, consistent with the literature).

(2) The primacy effect or recency effect observed in the experiments and traditional working memory models, including the slot model and the limited resource model, should be examined to see if it also appears in this model.

The literature on human working memory shows a prevalent recency effect (but not a primacy effect, which is thought to be due to episodic memory, and which is not included in our model). We have added a section showing that our model demonstrates clear recency effects.

(3) The construction of the model and the single neuron dynamics involved need further refinement and optimization:Model Description: The details of the model construction in the paper need to be further elaborated to help other researchers better understand and apply the model in reproducing or extending research. Specifically:a) The construction details of different modules in the model (such as Input signal, BG, striatum, superficial PFC, deep PFC) and the projection relationships between different modules. Adding a diagram to illustrate the network construction would be beneficial.

To aid in the understanding of the model construction and model components, we have included an additional figure (Figure 1: Base Model) that explains the key layers and components of the model. We have also altered the overall model figures to show more clearly that the inputs project to both PFC and striatum, to highlight that information is temporarily represented in superficial PFC layers even before striatal gating, which is needed for storage after the input decays.

We have expanded the methods and equations and we also provide a link to the model github for purposes of reproducibility and sharing.

A base model figure was added to specify key connections.

a) The numbers of excitatory and inhibitory neurons within different modules and the connections between neurons.

We added clarification on the type of connections between layers (specifying which are fixed and learned). We have also added the size of layers in a new appendix section “Layer Sizes and Inner Mechanics”

b) The dynamics of neurons in different modules need to be elaborated, including the description of the dynamic equations of variables (such as x) involved in single neuron equations.

Single neuron dynamics are explained in equations 1-4. Equations 5-6 explain how activation travels between layers. The specific inhibitory dynamics in the chunking layer are elaborated in Figure 4. PBWM Model and Chunking Layer Details. The Appendix section “Neural model implementational details” states the key equations, neural information and connectivity. Since there is a large corpus of background information underlying these models, we have linked the Emergent github and specifically the Computational Cognitive Neuroscience textbook which has a detailed description of all equations. For the sake of paper length and understability, we chose the most relevant equations that distinguish our model.

c) The selection of parameters in the model, especially those that significantly affect the model's performance.

The appendix section hyperparameter search details some of the key parameters and why those values were chosen.

d) The model employs a sequential working memory paradigm, the forms of external stimuli involved in the encoding and recalling phases (including their mathematical expressions, durations, strengths, and other parameters) need to be elaborated further.

We appreciate this comment. We have expanded the Appendix section “Continuous Stimuli” to include the details of stimuli presentation (including durations etc).

(4) The figures in the paper need optimization. For example, the size of the schematic diagram in Figure 2 needs to be enlarged, while the size of text such as "present stimulus 1, 2, recall stimulus 1" needs to be reduced. Additionally, the citation of figures in the main text needs to be standardized. For example, Figure 1b, Figure 1c, etc., are not cited in the main text.

The task sequence figure (original Figure 2) has been modified and following your suggestions, text sizes have been modified.

(5) Section 3.1 in the appendix is missing.

Supplemental section 3.1 is removed.